# Latent Space Chain-of-Embedding Enables Output-free LLM Self-Evaluation

**Yiming Wang**[α,†]  **Pei Zhang**[β,γ]  **Baosong Yang**[β,*]  **Derek F. Wong**[γ]  **Rui Wang**[α,*]

[α]Department of Computer Science and Engineering, Shanghai Jiao Tong University
{yiming.wang, wangrui12}@sjtu.edu.cn
[β]Tongyi Lab, Alibaba Group Inc.   [γ]NLP[2]CT Lab, University of Macau
yangbaosong.ybs@alibaba-inc.com

## Abstract

LLM self-evaluation relies on the LLM's own ability to estimate response correctness, which can greatly improve its deployment reliability. In this research track, we propose the **Chain-of-Embedding (CoE)** in the latent space to enable LLMs to perform output-free self-evaluation. CoE consists of all progressive hidden states produced during the inference time, which can be treated as the latent thinking path of LLMs. We find that when LLMs respond correctly and incorrectly, their CoE features differ, these discrepancies assist us in estimating LLM response correctness. Experiments in four diverse domains and seven LLMs fully demonstrate the effectiveness of our method. Meanwhile, its label-free design intent without any training and millisecond-level computational cost ensures real-time feedback in large-scale scenarios. More importantly, we provide interesting insights into LLM response correctness from the perspective of hidden state changes inside LLMs. The code is public at: https://github.com/Alsace08/Chain-of-Embedding.

## 1 Introduction

Large Language Models (LLMs) have excellent abilities to generalize across diverse scenarios (Achiam et al., 2023; Guo et al., 2025). However, their outputs can sometimes be unstable, leading to incorrect responses that may threaten social safety. Therefore, label-free LLM self-evaluation has emerged as a crucial research area, which estimates the correctness of LLM

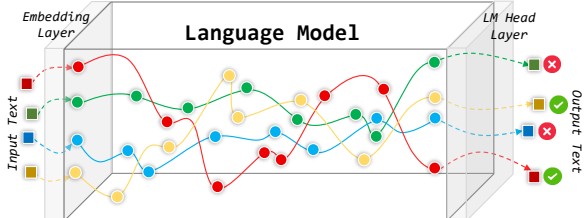

Figure 1: Chain-of-Embedding we propose in this paper mirrors the latent thinking path of LLMs, which may reflect LLM response correctness during inference time.

responses fully through LLMs' own capabilities. It can provide real-time response monitoring and feedback in large-scale employments, enhancing the reliability of LLMs (Sun et al., 2024).

Popular self-evaluation research in the era of LLMs focuses more on output-based forms (Zhang et al., 2023). Two typical paradigms that do not assess the internal states of LLMs involve directly asking LLMs to express confidence in their responses through well-designed prompts (Lin et al., 2022a; Tian et al., 2023), and generating multiple responses by perturbing prompts (Gao et al., 2024) or decoding sampling (Wang et al., 2023) to calculating the response consistency (Xiong et al., 2024). Besides the two types, other methods basically draw on uncertainty estimation concepts from the era of deep neural networks, leveraging output logits or probability distributions to gauge the confidence of model responses (Malinin & Gales, 2020; Si et al., 2022; Huang et al., 2023; Kuhn et al., 2023).

Recently, some research has revealed that the latent space of LLMs contains a substantial amount of untapped hidden state information, they can largely reflect response correctness (Azaria & Mitchell, 2023; Liu et al., 2023; Duan et al., 2024), and are usually more interpretable than LLM output (Li et al., 2024a). However, these output-free research often require correctness labels 0/1 for training probing classifiers to extract features from hidden states (Burns et al., 2022; Sky et al., 2024; Su et al., 2024). This contradicts our goal of being "label-free" and limits the generalization capabilities on unseen data. *To further expand this research line*, we consider a challenging but valuable question:

*How to solely utilize hidden states to estimate the LLM response correctness without any label?*

---

*Corresponding Authors. †Work done during Yiming's internship at Tongyi Lab, Alibaba Group Inc.

To answer this question, we start from the perspective of human thinking: In the cognitive theory, human thinking is accomplished collaboratively by intuitive thinking (system 1) and deliberative thinking (system 2) (Evans, 2003): correct thinking tends to activate system 2 to produce more deliberative thinking paths, while incorrect thinking tends to be affected by system 1 to make more rapid and direct thinking paths (Kahneman, 2011). This cognitive phenomenon means that *human thinking paths may differ when responding correctly and incorrectly.*

Next, we draw an analogy about the latent thinking path from humans to LLMs: Some research (Peters et al., 2018; Tenney, 2019; Jawahar et al., 2019; Chen et al., 2020) has demonstrated that large Transformer-based (Vaswani et al., 2017) language models generate text representations by first encoding morphological and syntactic information in the lower layers, then progressing to more complex semantic information in the higher layers. This means that the hidden state changes can mirror the interpretable progressive thinking of LLMs in the latent space. Moreover, some LLM mechanistic studies also utilize multiple hidden states to explore the thinking states at different stages of LLMs (Ye et al., 2024). Therefore, *we can treat the progressive hidden states as the latent thinking path of LLMs*, which we term **Chain-of-Embedding (CoE)**, as shown in Figure 1.

Based on these analyses, we boldly migrate human thinking patterns to LLMs and make the following assumption: ***CoE discrepancies may happen when LLMs generate correct and incorrect responses***. Starting from this assumption, our paper structure and contributions are as follows: In Section 2, we first explore the CoE discrepancy when LLMs respond correctly and incorrectly by quantifying its features to demonstrate this assumption. In Section 3, we propose a comprehensive CoE-based metric for label-free LLM self-evaluation. In Section 4, we verify the performance of our CoE method from four diverse domains that are popular in the LLM ability test: *Mathematics, Reasoning, Knowledge, and Understanding*, to demonstrate the effectiveness of CoE for self-evaluation. In Section 5, we conduct theoretical analyses further to present more insights about the effectiveness of our method.

**Problem Statement.** *Label-free LLM self-evaluation aims to estimate whether the LLM response to a given input question is correct or not **fully through LLMs' own capabilities without relying on any true label, external tool, and supervised trainer*** (Chen & Mueller, 2023; Li et al., 2024c). We denote the language model as $f$. For each sample $(\boldsymbol{x}, \boldsymbol{y}) \leftrightarrow$ (question, true answer), the output of the language model is denoted as $\hat{\boldsymbol{y}} = f(\boldsymbol{x})$. Then this sample is associated with a decision score $s = S(\boldsymbol{x}, \hat{\boldsymbol{y}}, f)$, where $S(\cdot)$ is a decision function derived solely from the question, LLM output, and language model information — without reference to the true label $\boldsymbol{y}$. The domain of this function encompasses the entire sample and model space. An ideal decision function aims to achieve the following goal: ***a higher score indicates a greater likelihood of a correct answer***. Therefore, the self-evaluation task can be formulated as a binary classification problem. Let $\gamma$ be the decision threshold, for a question, whether the LLM response is correct can be discriminated as the instruction function $\chi(\hat{\boldsymbol{y}}) = $ Correct if $s > \gamma$ else Incorrect. The core goal is to find an optimal threshold to improve the classification accuracy.

## 2 Chain-of-Embedding Reflects Response Correctness

### 2.1 Definition: Latent Space Chain-of-Embedding (CoE)

**Formalization.** We start by formalizing the CoE under the language model $f$. Assume the model has $L$ hidden layers, we can decompose $f$ into the following ordered sub-modules:

$$f = f_{\text{head}} \circ f_L \circ \cdots \circ f_l \circ \cdots \circ f_1 \circ f_{\text{emb}}. \tag{1}$$

In Eq.1, $f_{\text{head}} : \mathbb{R}^d \to \mathbb{R}^{|\mathcal{V}|}$ is the final classification layer, $f_{\text{emb}} : \mathbb{R}^{|\mathcal{V}|} \to \mathbb{R}^d$, which can also be denoted as $f_0$, is the embedding mapping layer (0-th layer), and each $f_l(1 \le l \le L) : \mathbb{R}^d \to \mathbb{R}^d$ is the intermediate hidden layer. Here $d$ is the embedding dimensions and $\mathcal{V}$ is the model vocabulary.

Given a question $\boldsymbol{x}$ as input to $f$, the output $\hat{\boldsymbol{y}}$ consists of $T$ tokens $\hat{y}_1 \hat{y}_2 ... \hat{y}_T$. For the $t$-th token, we denote its hidden state at layer $l(0 \le l \le L)$, *i.e.*, the $t$-th output embedding of function $f_l \circ \cdots \circ f_1 \circ f_0$, as $\boldsymbol{z}_l^t$. Following the definitions in Ren et al. (2022); Wang et al. (2024a), we define the average embedding at layer $l$ as $\boldsymbol{h}_l = \frac{1}{T} \sum_{t=1}^{T} \boldsymbol{z}_l^t$, which represents the $l$-th sentence hidden state. Then, the CoE is expressed as a progressive chain $\boldsymbol{H}$ of all sentence hidden states formalized as follows:

$$\boldsymbol{H} = \underbrace{\boldsymbol{h}_0}_{\text{Input State}} \to \underbrace{\boldsymbol{h}_1 \to \cdots \to \boldsymbol{h}_l \to \cdots \to \boldsymbol{h}_{L-1}}_{\text{Intermediate Hidden States}} \to \underbrace{\boldsymbol{h}_L}_{\text{Output State}} \qquad (\textbf{Chain-of-Embedding})$$

**Feature Definition.** After formalizing the CoE, we need to quantify its features so that we can utilize them to validate the assumption proposed in Section 1: "*CoE discrepancies may happen when LLMs generate correct and incorrect responses*". We can create a continuous CoE trajectory in the latent space by performing segmented linear interpolation (simply connecting adjacent states) on the CoE. To measure the trajectory feature, its geometric information is the most fundamental dimension (Helland-Hansen & Hampson, 2009; Rintoul & Wilson, 2015), which usually includes **magnitude** and **angle** that can reflect the distance and direction changes produced during the trajectory wandering.

We briefly examine the practical significance of these two features in measuring the LLMs' thinking path. The magnitude feature is undoubtedly the direct feature of the thinking path curvature. In contrast, while the angle feature does not explicitly represent the thinking path curvature, the cosine value between two embeddings indicates semantic similarity (Rahutomo et al., 2012). This suggests that the angle feature can indirectly reflect the thinking path feature at the semantic modeling level.

Now, we first define changes in magnitude and angle between each adjacent state pair $(\boldsymbol{h}_l, \boldsymbol{h}_{l+1})(0 \leq l \leq L - 1)$. The magnitude change $M(\boldsymbol{h}_l, \boldsymbol{h}_{l+1})$ is quantified using the L2-norm, while the angle change $A(\boldsymbol{h}_l, \boldsymbol{h}_{l+1})$ is derived by indirectly calculating the cosine value between the two vectors, and then transformed using the arc cosine function. The two measures are formalized as follows:

$$M(\boldsymbol{h}_l, \boldsymbol{h}_{l+1}) = ||\boldsymbol{h}_{l+1} - \boldsymbol{h}_l||_2, \quad A(\boldsymbol{h}_l, \boldsymbol{h}_{l+1}) = \arccos\left(\frac{\boldsymbol{h}_{l+1}^{\top}\boldsymbol{h}_l}{||\boldsymbol{h}_{l+1}||_2 \cdot ||\boldsymbol{h}_l||_2}\right). \quad (2)$$

Subsequently, the magnitude and angle features of the whole CoE trajectory, denoted as $\mathrm{Mag}(\boldsymbol{H})$ and $\mathrm{Ang}(\boldsymbol{H})$, can be defined as the average changes in magnitude and angle between each adjacent state pair. The two features are formalized as follows:

$$\mathrm{Mag}(\boldsymbol{H}) = \frac{1}{L} \cdot \sum_{l=0}^{L-1} \frac{M(\boldsymbol{h}_l, \boldsymbol{h}_{l+1})}{\mathcal{Z}_{\mathrm{Mag}}}, \quad \mathrm{Ang}(\boldsymbol{H}) = \frac{1}{L} \cdot \sum_{l=0}^{L-1} \frac{A(\boldsymbol{h}_l, \boldsymbol{h}_{l+1})}{\mathcal{Z}_{\mathrm{Ang}}}. \quad (3)$$

In Eq.3, to reduce potential sample biases, we set the range scaling factors $\mathcal{Z}_{\mathrm{Mag}} = M(\boldsymbol{h}_0, \boldsymbol{h}_L)$ and $\mathcal{Z}_{\mathrm{Ang}} = A(\boldsymbol{h}_0, \boldsymbol{h}_L)$ for the following reason: The positions in the input space and output space of different samples may vary, if the input and output of one sample are far apart in the latent space, its trajectory naturally has a longer wandering distance. By setting scaling factors, we convert the absolute magnitude and angle changes of each adjacent state pair into relative changes, specifically, the changes of $(\boldsymbol{h}_l, \boldsymbol{h}_{l+1})$ is relative to the changes of the input-output states $(\boldsymbol{h}_0, \boldsymbol{h}_L)$, thereby avoiding measurement noise caused by inherent differences between samples.

## 2.2 How CoE Reflects Response Correctness

**Setup.** After defining CoE and its features, we aim to explore the generalized impact of CoE on the LLM response correctness. Therefore, we focus on four popular domains: *Mathematics*, *Reasoning*, *Knowledge*, *Understanding*, **with each domain set MATH, TheoremQA, MMLU, and Belebele as domain datasets** separately. Dataset details and citations can be found in Section 4.1. In each domain dataset, we divide the correct and incorrect samples into two sets, with each corresponding to a feature set, denoted as $V_+ = \{(\mathrm{Mag}_i^+, \mathrm{Ang}_i^+)\}_{i=1}^{n_+}$ and $V_- = \{(\mathrm{Mag}_i^-, \mathrm{Ang}_i^-)\}_{i=1}^{n_-}$. Additionally, we use the **Qwen2-7B-Instruct** (Yang et al., 2024) model as the backbone. These settings apply to all experimental analyses in this section (Figure 2 and 3) and will not be repeated hereafter.

**CoE Feature Distribution Discrepancy.** Now, we quantify the CoE feature discrepancies between correct and incorrect sample sets for each domain. Due to each sample having two CoE features, we employ 2D kernel density estimation to calculate the probability density function (PDF) $f_V$ for each feature set $V = \{(\mathrm{Mag}_i, \mathrm{Ang}_i)\}_{i=1}^{n}$, which represent the CoE feature distribution of $V$. We use Gaussian kernel for PDF estimation, expressed as:

$$f_V(\mathrm{Mag}, \mathrm{Ang}) = \frac{1}{nh^2} \sum_{i=1}^{n} \frac{1}{2\pi} \exp\left\{-\frac{1}{2h^2}[(\mathrm{Mag} - \mathrm{Mag}_i)^2 + (\mathrm{Ang} - \mathrm{Ang}_i)^2]\right\}, \quad (4)$$

where $h$ is the bandwidth, which we set to 1 to maintain consistency with the default value in the Python `sklearn` library (Pedregosa et al., 2011). We implement Eq.4 using the `sklearn` library. In each domain, we independently derive the PDFs $f_{V_+}$ and $f_{V_-}$ for $V_+$ and $V_-$ using Eq.4, as illustrated in Figure 2. We find that in all domains, the distributions of correct and incorrect samples do not overlap and show significant discrepancies, with conclusions described below:

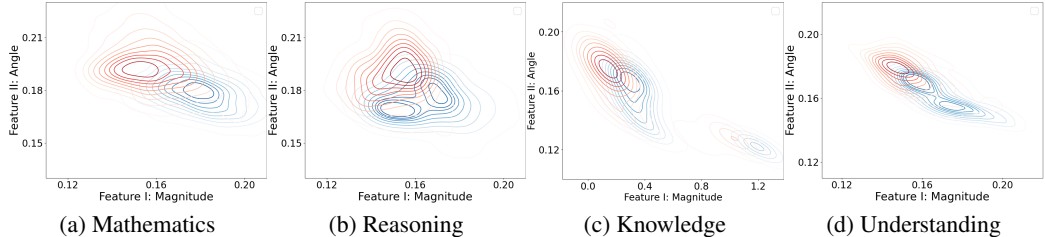

(a) Mathematics  (b) Reasoning  (c) Knowledge  (d) Understanding

Figure 2: **CoE feature distribution** of correct and incorrect sample sets in four diverse domains. Blue and Red distributions represent the correct samples and incorrect samples, respectively. Datasets and the model used in this figure are shown in Section 2.2 "Setup".

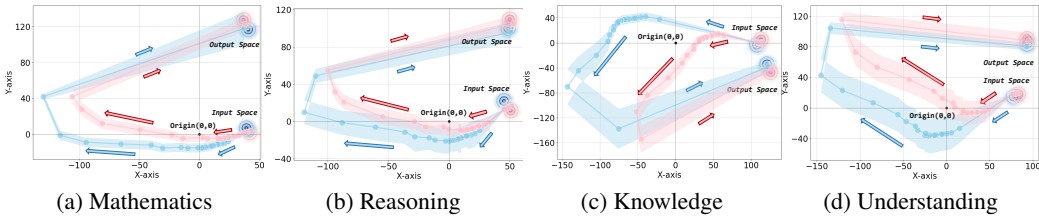

(a) Mathematics  (b) Reasoning  (c) Knowledge  (d) Understanding

Figure 3: **CoE trajectory visualization** of correct and incorrect sample sets in four diverse domains. Blue and Red distributions represent the correct samples and incorrect samples, respectively. Each trajectory represents the average trajectories of all correct or incorrect samples in corresponding datasets, and the shades represent the trajectory standard variances of all samples. We project the CoE into two-dimensional space using PCA for dimensionality reduction, which maintains the relative positioning of the data as constant as possible by rotating the coordinate system (Dunteman, 1989). Datasets and the model used in this figure are shown in Section 2.2 "Setup".

- *CoE magnitude feature* Mag of correct samples are more significant than that of incorrect samples, indicating that LLMs' latent thinking paths are more convoluted when providing correct answers.
- *CoE angle feature* Ang of correct samples are less significant than that of incorrect samples, indicating that LLMs' latent thinking paths at the semantic modeling level are more unstable when providing incorrect answers.

**CoE Visualization.** These two conclusions directly reveal the essential discrepancies of CoE features. To demonstrate these more intuitively, we visualize CoE as illustrated in Figure 3: Overall, the CoE trajectory is not inclined to choose the shortest path from the input space to the output space; instead, it passes through a semantic space that is quite distant from both the input and output. In detail, the detour phenomenon of the CoE trajectory for correct samples is more pronounced than that for incorrect samples, which verifies its more significant CoE magnitude feature. Furthermore, the intermediate states of the incorrect samples are closer to the origin, resulting in a larger angle formed before and after the state transition, which verifies its more significant CoE angle feature.

## 3 CoE Score for Output-free LLM Self-evaluation

In Section 2, we have quantified CoE features and highlighted discrepancies between correct and incorrect samples. Next, we wish to create a self-evaluation metric combining the two features, so as to detect the LLM response correctness using a comprehensive CoE feature. However, the feature combination is not simple because of their inconsistent magnitudes. We propose two ways as follows:

**CoE-R: Real-Space Combination.** A straightforward method is to calculate a numerical summation of the two features. Though adding the two real numbers may seem unmeaningful due to their differing magnitudes, our focus is on the metric relative trends rather than exact numbers, so this way can preserve the metric usability without considering the feature relevance.

In Section 2.2, we have found that $\mathrm{Ang}(\boldsymbol{H})$ is inversely proportional to the response correctness, so for each adjacent state pair, we use $1 - A(\boldsymbol{h}_l, \boldsymbol{h}_{l+1})/A(\boldsymbol{h}_0, \boldsymbol{h}_L)$ as the direction change measure, then add it with $M(\boldsymbol{h}_l, \boldsymbol{h}_{l+1})/M(\boldsymbol{h}_0, \boldsymbol{h}_L)$. By removing extraneous constants, we derive CoE-R($\boldsymbol{H}$) score by averaging all adjacent state pair changes with the real-space combination way:

$$\mathrm{CoE\text{-}R}(\boldsymbol{H}) = \frac{1}{L} \cdot \sum_{l=0}^{L-1} \left( \frac{M(\boldsymbol{h}_l, \boldsymbol{h}_{l+1})}{M(\boldsymbol{h}_0, \boldsymbol{h}_L)} - \frac{A(\boldsymbol{h}_l, \boldsymbol{h}_{l+1})}{A(\boldsymbol{h}_0, \boldsymbol{h}_L)} \right). \tag{5}$$

**CoE-C: Complex-Space Combination.** However, while the exact values of the two features may not be critical, they can significantly interfere with each other, particularly if one is abnormally large. This interference can weaken the overall impact of another feature, resulting in the instability of the CoE-R metric. Therefore, we aim to combine the two features more seamlessly.

The magnitude and angle features enable a clear association of complex numbers in the complex plane, with each point uniquely represented by its complex magnitude and complex argument. Therefore, for each adjacent state pair, We combine $M(\boldsymbol{h}_l, \boldsymbol{h}_{l+1})$ and $A(\boldsymbol{h}_l, \boldsymbol{h}_{l+1})$ into a new feature point $C(\boldsymbol{h}_l, \boldsymbol{h}_{l+1})$ on the complex plane, where $M(\boldsymbol{h}_l, \boldsymbol{h}_{l+1})$ represents the complex magnitude and $A(\boldsymbol{h}_l, \boldsymbol{h}_{l+1})$ represents the complex argument. This feature point $C(\boldsymbol{h}_l, \boldsymbol{h}_{l+1})$ can be expressed as:

$$
\begin{aligned}
C(\boldsymbol{h}_l, \boldsymbol{h}_{l+1}) &= M(\boldsymbol{h}_l, \boldsymbol{h}_{l+1})e^{i \cdot A(\boldsymbol{h}_l, \boldsymbol{h}_{l+1})} \\
&= \underbrace{M(\boldsymbol{h}_l, \boldsymbol{h}_{l+1})\cos(A(\boldsymbol{h}_l, \boldsymbol{h}_{l+1}))}_{\text{Real Part: } \mathrm{Re}(C(\boldsymbol{h}_l, \boldsymbol{h}_{l+1}))} + i \cdot \underbrace{M(\boldsymbol{h}_l, \boldsymbol{h}_{l+1})\sin(A(\boldsymbol{h}_l, \boldsymbol{h}_{l+1}))}_{\text{Imaginary Part: } \mathrm{Im}(C(\boldsymbol{h}_l, \boldsymbol{h}_{l+1}))},
\end{aligned}
\tag{6}
$$

where $i$ is the imaginary unit. Each adjacent state pair corresponds to one feature point, we then average these $L$ feature points by separately averaging their real and imaginary parts. The magnitude of this averaged point yields the final CoE-C($\boldsymbol{H}$) score with the complex-space combination way:

$$
\begin{aligned}
\text{CoE-C}(\boldsymbol{H}) &= \sqrt{\left(\frac{1}{L} \cdot \sum_{l=0}^{L-1} \mathrm{Re}(C(\boldsymbol{h}_l, \boldsymbol{h}_{l+1}))\right)^2 + \left(\frac{1}{L} \cdot \sum_{l=0}^{L-1} \mathrm{Im}(C(\boldsymbol{h}_l, \boldsymbol{h}_{l+1}))\right)^2} \\
&= \frac{1}{L} \cdot \sqrt{\left(\sum_{l=0}^{L-1} M(\boldsymbol{h}_l, \boldsymbol{h}_{l+1})\cos(A(\boldsymbol{h}_l, \boldsymbol{h}_{l+1}))\right)^2 + \left(\sum_{l=0}^{L-1} M(\boldsymbol{h}_l, \boldsymbol{h}_{l+1})\sin(A(\boldsymbol{h}_l, \boldsymbol{h}_{l+1}))\right)^2}.
\end{aligned}
\tag{7}
$$

This combination metric still tends to capture relative trends, while fundamentally aiming to establish more robust mutual constraints between the two features, in contrast to CoE-R. First, the monotonicity of CoE-C and CoE-R remains consistent and does not contradict the conclusions of Section 2.2. Second, CoE-C exhibits less variation than CoE-R when confronted with outliers, allowing for fewer deviations from the feature distribution of the corresponding class. See Section 5 for concrete proofs.

Figure 4 shows the **Computation Sketch** of the two CoE scores, and the complete **Algorithmic Process** of the two CoE scores is shown in Appendix B.1. Higher scores indicate a greater likelihood of obtaining the correct response. In terms of **Computational Complexity**, after the base LLM inference and the extraction of all hidden states, the computations for both CoE-R and CoE-C involve only $\mathcal{O}(Ld)$ operations for scalar addition and multiplication, along with $\mathcal{O}(L)$ operations for square roots and trigonometric computations. Since these computations can be executed in parallel by the CPU, the overall computational burden is negligible. Section 4.3 will provide a detailed efficiency analysis of the execution time in practice.

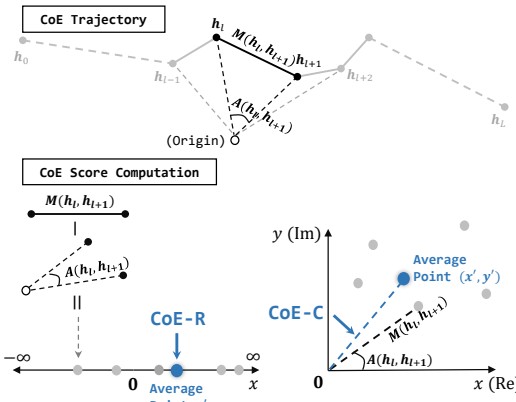

Figure 4: Sketch of CoE Score computation.

## 4 EXPERIMENTAL VERIFICATION

### 4.1 SETUP

**Dataset.** We select six datasets across four domains for our self-evaluation experiments. These domains reflect the four critical dimensions of LLM capabilities (Zheng et al., 2024; Huang et al., 2024): (1) **GSM8K** (Cobbe et al., 2021) and **MATH** (Hendrycks et al., 2021) for the *Mathematics* domain; (2) **CommonsenseQA** (Talmor et al., 2019) and **TheoremQA** (Chen et al., 2023) for the *Reasoning* domain; (3) **MMLU** (Hendrycks et al., 2020) for the *Knowledge* domain; (4) **Belebele** (Bandarkar et al., 2023) for the *Understanding* domain. Dataset details are shown in Appendix C.1.

**Language Model.** We use *instruction-based* models due to their ability to follow instructions, effectively addressing diverse user needs. We mainly adopt 7B+ parameter models with the Zero-Shot-CoT generation paradigm (Wei et al., 2022; Kojima et al., 2022), including **Llama2-7B** (Touvron

Table 1: AUROC, FPR95, and AUPR results of all methods in four diverse domains with different LLMs. *Italics* means that this method does not assess internal states, "†" means that this method requires multiple stochastic inferences for sampling multiple outputs, and "*" means that this method is output-free and only utilizes hidden states. Additionally, underline represents the SOTA performance among all baselines, **bold** represents the SOTA performance among all methods.

| Method / Model | Llama2-7B-Instruct | Llama3-8B-Instruct | Qwen1.5-7B-Instruct | Qwen2-7B-Instruct | Mistral-7B-Instruct | Llama3-70B-Instruct | Qwen2-72B-Instruct | Average |
|---|---|---|---|---|---|---|---|---|
| **Domain I: Mathematics (AUROC ↑ / FPR95 ↓ / AUPR ↑)** | | | | | | | | |
| 1. *Verbal Conf.* | 41.08 / 99.82 / 12.26 | 44.19 / 98.94 / 43.56 | 56.69 / 89.73 / 67.75 | 56.21 / 91.42 / 68.23 | 43.26 / 97.16 / 27.85 | 61.07 / 88.80 / 77.43 | 63.34 / 81.37 / 70.05 | 52.26 / 92.46 / 52.44 |
| 2. *PSA pipeline* † | 52.51 / 95.94 / 14.74 | 61.34 / 86.24 / 51.24 | 68.73 / 80.28 / 50.83 | 62.28 / 85.53 / 71.15 | 75.68 / 79.26 / 44.98 | 67.02 / 88.19 / 78.40 | 70.22 / 75.40 / 74.09 | 65.39 / 84.40 / 55.06 |
| 3. Max Prob. | 54.61 / 91.99 / 19.24 | 58.47 / 84.60 / 54.81 | 71.09 / 77.60 / 55.08 | 68.40 / 75.67 / 73.82 | 65.53 / 81.09 / 37.33 | 66.70 / 87.25 / 79.70 | 70.16 / 71.68 / 74.47 | 64.99 / 81.41 / 56.35 |
| 4. Perplexity | 54.81 / 92.49 / 18.97 | 58.32 / 84.71 / 54.72 | 71.68 / 76.20 / 55.42 | 68.83 / 73.09 / 74.23 | 66.37 / 79.40 / 37.93 | 66.59 / 86.30 / 79.97 | 70.80 / 71.47 / 74.16 | 65.34 / 80.52 / 56.48 |
| 5. Entropy | 54.93 / 93.00 / 19.55 | 60.17 / 84.02 / 55.57 | 72.29 / 74.04 / 56.06 | 70.25 / 73.45 / 74.97 | 66.55 / 82.02 / 38.27 | 68.03 / 86.12 / 80.16 | 70.40 / 65.82 / 74.36 | 66.08 / 79.78 / 56.99 |
| 6. Temp. Scaling | 54.36 / 92.44 / 19.43 | 58.72 / 84.33 / 54.77 | 70.95 / 80.03 / 53.24 | 69.11 / 76.98 / 73.12 | 66.37 / 79.40 / 37.03 | 66.24 / 87.71 / 79.30 | 70.23 / 76.39 / 73.82 | 64.94 / 82.90 / 55.81 |
| 7. Energy | 48.76 / 96.88 / 11.07 | 53.26 / 92.64 / 50.32 | 54.12 / 94.71 / 42.86 | 56.10 / 88.52 / 70.76 | 49.07 / 96.57 / 26.94 | 51.01 / 95.73 / 69.58 | 57.20 / 87.42 / 66.25 | 52.78 / 93.21 / 48.25 |
| 8. MC Dropout † | 51.35 / 96.55 / 14.28 | 48.75 / 96.25 / 47.18 | 52.68 / 92.84 / 40.71 | 55.64 / 90.53 / 68.58 | 49.09 / 94.12 / 27.26 | 53.67 / 92.28 / 72.41 | 54.80 / 90.46 / 62.25 | 52.28 / 93.29 / 47.52 |
| 9. LN-Entropy † | 56.74 / 89.96 / 19.07 | 61.95 / 85.61 / 56.63 | 67.40 / 79.24 / 50.34 | 66.63 / 71.09 / 73.84 | 67.71 / 80.16 / 39.25 | 68.72 / 85.48 / 80.97 | 68.39 / 61.19 / 73.62 | 65.36 / 78.96 / 56.33 |
| 10. EigenScore †* | 44.84 / 97.04 / 16.58 | 52.77 / 95.45 / 43.00 | 35.55 / 100.00 / 31.23 | 62.56 / 88.89 / 67.10 | 41.65 / 96.77 / 24.11 | 57.61 / 91.07 / 75.52 | 40.02 / 99.21 / 58.54 | 47.85 / 95.49 / 45.15 |
| CoE-R (Ours) * | 64.23 / 84.47 / 20.94 | 72.54 / 75.61 / 66.96 | 38.44 / 96.78 / 33.60 | 75.75 / 65.95 / 83.67 | 70.77 / 82.66 / 42.68 | 79.61 / 67.01 / 84.19 | 84.30 / 59.32 / 82.37 | 69.38 / 75.97 / 59.20 |
| CoE-C (Ours) * | 63.63 / 85.01 / 20.67 | 73.08 / 79.60 / 68.99 | 77.22 / 78.44 / 58.23 | 76.68 / 64.48 / 84.57 | 72.24 / 77.11 / 43.55 | 79.35 / 70.63 / 84.30 | 84.34 / 53.25 / 83.18 | 75.22 / 72.64 / 63.36 |
| **Domain II: Reasoning (AUROC ↑ / FPR95 ↓ / AUPR ↑)** | | | | | | | | |
| 1. *Verbal Conf.* | 54.17 / 95.88 / 29.54 | 50.01 / 97.74 / 45.62 | 52.70 / 96.23 / 51.78 | 43.25 / 99.15 / 42.73 | 44.83 / 96.71 / 36.51 | 42.67 / 99.81 / 49.27 | 46.63 / 97.93 / 42.20 |
| 2. *PSA pipeline* † | 52.10 / 89.25 / 28.50 | 51.25 / 94.76 / 47.08 | 62.28 / 94.96 / 61.63 | 48.98 / 97.59 / 49.53 | 46.07 / 97.30 / 38.84 | 51.64 / 96.55 / 52.87 | 48.83 / 93.49 / 60.71 | 51.59 / 94.84 / 48.45 |
| 3. Max Prob. | 55.68 / 92.82 / 32.05 | 48.14 / 95.10 / 44.43 | 63.64 / 95.00 / 60.99 | 54.57 / 91.77 / 54.21 | 53.90 / 91.50 / 39.53 | 49.82 / 94.61 / 56.10 | 56.36 / 91.45 / 65.45 | 54.58 / 93.18 / 50.39 |
| 4. Perplexity | 55.78 / 92.11 / 32.26 | 48.40 / 95.62 / 44.53 | 63.94 / 93.60 / 61.23 | 55.39 / 93.33 / 54.78 | 54.70 / 90.84 / 40.02 | 50.36 / 94.22 / 56.41 | 56.66 / 90.84 / 65.83 | 55.03 / 92.94 / 50.72 |
| 5. Entropy | 55.55 / 93.03 / 31.05 | 48.56 / 95.55 / 44.64 | 63.99 / 93.74 / 60.96 | 55.97 / 92.69 / 54.80 | 54.83 / 90.04 / 40.12 | 50.61 / 94.04 / 56.52 | 56.74 / 90.99 / 65.62 | 55.18 / 92.87 / 50.53 |
| 6. Temp. Scaling | 55.52 / 93.32 / 31.76 | 48.12 / 95.47 / 44.39 | 63.38 / 94.17 / 60.76 | 54.06 / 94.04 / 53.87 | 53.43 / 92.23 / 39.23 | 49.46 / 95.07 / 55.97 | 56.18 / 91.97 / 65.06 | 54.30 / 93.75 / 50.14 |
| 7. Energy | 49.74 / 96.23 / 26.57 | 43.69 / 97.02 / 41.29 | 52.18 / 96.66 / 53.20 | 47.52 / 95.16 / 50.07 | 48.34 / 95.61 / 36.84 | 44.49 / 98.04 / 50.05 | 51.35 / 94.25 / 58.72 | 48.18 / 96.14 / 45.24 |
| 8. MC Dropout † | 50.46 / 95.53 / 28.74 | 46.21 / 96.60 / 42.07 | 51.18 / 97.24 / 52.69 | 50.14 / 97.17 / 49.86 | 51.29 / 95.09 / 37.44 | 48.97 / 95.39 / 52.22 | 52.17 / 95.25 / 59.53 | 50.06 / 96.04 / 46.07 |
| 9. LN-Entropy † | 51.26 / 97.42 / 32.87 | 52.88 / 95.36 / 40.92 | 59.42 / 94.48 / 57.14 | 56.07 / 94.25 / 54.26 | 59.74 / 92.26 / 44.77 | 48.26 / 97.78 / 53.08 | 54.44 / 95.16 / 60.20 | 54.58 / 95.24 / 49.03 |
| 10. EigenScore †* | 47.01 / 95.46 / 26.11 | 53.58 / 95.94 / 45.80 | 47.78 / 97.96 / 45.68 | 53.39 / 94.11 / 53.39 | 60.70 / 91.12 / 48.29 | 43.59 / 99.86 / 45.38 | 57.24 / 91.20 / 65.99 | 51.89 / 95.09 / 47.23 |
| CoE-R (Ours) * | 55.51 / 88.40 / 32.76 | 63.12 / 89.83 / 54.68 | 58.19 / 93.28 / 57.10 | 66.68 / 85.84 / 64.41 | 72.62 / 89.01 / 56.82 | 63.90 / 87.11 / 65.53 | 62.54 / 89.77 / 68.46 | 63.22 / 89.03 / 57.05 |
| CoE-C (Ours) * | 59.00 / 86.69 / 34.36 | 55.85 / 90.14 / 50.18 | 67.67 / 86.44 / 63.10 | 62.70 / 87.42 / 62.91 | 70.79 / 88.97 / 55.31 | 66.93 / 85.53 / 68.31 | 61.86 / 90.99 / 68.38 | 63.54 / 88.02 / 57.51 |
| **Domain III: Knowledge (AUROC ↑ / FPR95 ↓ / AUPR ↑)** | | | | | | | | |
| 1. *Verbal Conf.* | 43.36 / 99.86 / 26.41 | 46.37 / 99.56 / 59.34 | 42.49 / 99.16 / 43.82 | 51.25 / 96.61 / 49.62 | 53.52 / 92.78 / 51.75 | 51.16 / 97.34 / 96.93 | 45.20 / 98.94 / 54.41 | 47.62 / 97.75 / 54.61 |
| 2. *PSA pipeline* † | 53.21 / 92.14 / 37.89 | 54.95 / 86.34 / 69.64 | 58.92 / 87.60 / 62.80 | 60.25 / 88.81 / 71.70 | 50.78 / 95.64 / 55.49 | 52.37 / 96.68 / 82.57 | 56.11 / 91.01 / 65.09 | 56.11 / 91.01 / 65.09 |
| 3. Max Prob. | 48.75 / 96.21 / 33.64 | 49.92 / 93.99 / 66.13 | 61.33 / 87.60 / 63.50 | 57.09 / 95.31 / 71.76 | 53.15 / 96.28 / 52.42 | 46.39 / 92.80 / 74.83 | 64.81 / 88.66 / 89.18 | 54.49 / 92.98 / 64.49 |
| 4. Perplexity | 49.70 / 95.66 / 34.19 | 50.50 / 92.90 / 66.59 | 61.04 / 88.20 / 63.71 | 57.26 / 94.79 / 72.04 | 53.41 / 95.95 / 52.95 | 46.65 / 93.60 / 75.20 | 64.80 / 89.69 / 89.31 | 54.76 / 92.83 / 64.85 |
| 5. Entropy | 49.11 / 95.93 / 33.79 | 50.12 / 92.35 / 66.25 | 60.40 / 87.40 / 62.99 | 57.80 / 93.75 / 71.84 | 54.39 / 94.59 / 53.42 | 46.07 / 90.40 / 74.85 | 66.10 / 89.69 / 89.50 | 54.85 / 91.87 / 64.66 |
| 6. Temp. Scaling | 48.13 / 97.02 / 33.28 | 49.52 / 93.99 / 65.84 | 61.47 / 87.60 / 63.48 | 56.81 / 93.75 / 71.40 | 53.18 / 96.96 / 52.14 | 47.74 / 91.33 / 76.90 | 66.01 / 87.42 / 89.32 | 54.69 / 92.58 / 64.62 |
| 7. Energy | 45.40 / 98.46 / 29.57 | 46.90 / 95.77 / 61.74 | 50.08 / 96.18 / 55.87 | 48.73 / 97.72 / 64.79 | 46.58 / 99.08 / 45.73 | 42.54 / 95.63 / 68.62 | 53.92 / 96.14 / 81.69 | 47.73 / 97.00 / 58.28 |
| 8. MC Dropout † | 46.56 / 98.68 / 31.12 | 49.87 / 93.16 / 65.79 | 56.78 / 91.24 / 59.09 | 55.64 / 95.88 / 68.78 | 50.24 / 98.43 / 50.69 | 44.68 / 97.59 / 70.50 | 53.45 / 94.32 / 82.65 | 51.03 / 95.61 / 61.25 |
| 9. LN-Entropy † | 42.69 / 99.64 / 28.51 | 55.36 / 91.14 / 68.90 | 54.62 / 92.25 / 57.43 | 58.01 / 92.68 / 71.99 | 60.42 / 90.36 / 57.63 | 53.09 / 90.05 / 76.84 | 45.78 / 96.25 / 77.88 | 52.85 / 93.20 / 62.74 |
| 10. EigenScore †* | 51.96 / 95.39 / 36.58 | 58.30 / 91.98 / 69.73 | 50.86 / 94.80 / 58.47 | 50.36 / 96.35 / 67.53 | 61.68 / 89.53 / 62.75 | 37.90 / 99.94 / 61.02 | 56.39 / 90.15 / 86.64 | 52.49 / 93.99 / 63.24 |
| CoE-R (Ours) * | 62.76 / 85.80 / 45.19 | 64.20 / 83.06 / 77.70 | 49.48 / 97.20 / 55.79 | 63.14 / 87.50 / 74.01 | 64.03 / 88.85 / 61.08 | 70.13 / 84.00 / 87.29 | 72.15 / 78.35 / 92.16 | 63.70 / 86.39 / 70.46 |
| CoE-C (Ours) * | 59.07 / 87.97 / 39.36 | 62.45 / 88.03 / 75.99 | 62.11 / 87.20 / 68.94 | 61.85 / 92.19 / 73.57 | 62.18 / 86.49 / 60.90 | 66.41 / 84.00 / 84.18 | 73.15 / 78.35 / 92.46 | 63.89 / 85.21 / 70.77 |
| **Domain IV: Understanding (AUROC ↑ / FPR95 ↓ / AUPR ↑)** | | | | | | | | |
| 1. *Verbal Conf.* | 47.23 / 99.59 / 32.86 | 47.42 / 99.10 / 83.64 | 46.85 / 98.27 / 73.38 | 51.18 / 95.67 / 79.65 | 50.21 / 97.33 / 45.24 | 49.82 / 98.46 / 87.20 | 60.04 / 87.15 / 92.06 | 50.39 / 96.51 / 70.57 |
| 2. *PSA pipeline* † | 56.35 / 91.23 / 54.54 | 52.07 / 89.64 / 89.37 | 57.11 / 90.61 / 85.04 | 62.07 / 93.20 / 92.15 | 54.19 / 94.88 / 50.82 | 60.65 / 88.25 / 97.13 | 70.49 / 86.67 / 94.43 | 58.99 / 90.64 / 80.49 |
| 3. Max Prob. | 48.95 / 97.31 / 46.40 | 56.64 / 95.19 / 90.42 | 44.96 / 93.92 / 77.36 | 60.52 / 88.35 / 91.80 | 60.82 / 92.32 / 56.70 | 60.12 / 89.74 / 97.09 | 76.41 / 76.19 / 98.05 | 58.34 / 90.43 / 79.68 |
| 4. Perplexity | 49.09 / 97.52 / 46.45 | 56.64 / 94.23 / 90.44 | 46.07 / 95.03 / 78.07 | 60.93 / 87.38 / 91.80 | 61.38 / 91.92 / 57.50 | 59.21 / 92.31 / 97.04 | 77.28 / 76.19 / 97.99 | 58.65 / 90.65 / 79.89 |
| 5. Entropy | 47.81 / 98.14 / 45.55 | 56.78 / 92.31 / 90.44 | 46.09 / 92.27 / 77.22 | 62.65 / 86.41 / 92.13 | 61.87 / 91.11 / 58.11 | 59.97 / 92.31 / 97.08 | 77.56 / 71.43 / 98.28 | 58.96 / 89.14 / 79.82 |
| 6. Temp. Scaling | 48.65 / 97.72 / 46.17 | 56.63 / 95.19 / 90.38 | 44.26 / 94.48 / 76.83 | 60.45 / 88.35 / 91.84 | 60.35 / 92.12 / 56.07 | 58.64 / 92.58 / 96.42 | 76.69 / 75.59 / 97.92 | 57.95 / 90.86 / 79.37 |
| 7. Energy | 45.17 / 97.00 / 47.40 | 44.78 / 98.06 / 80.14 | 42.09 / 96.57 / 72.34 | 53.26 / 97.38 / 81.67 | 50.78 / 95.64 / 50.22 | 52.15 / 99.25 / 92.06 | 63.44 / 85.11 / 93.41 | 51.08 / 95.57 / 73.89 |
| 8. MC Dropout † | 45.87 / 99.02 / 42.33 | 51.20 / 97.89 / 85.68 | 47.89 / 94.23 / 77.43 | 58.23 / 91.42 / 87.09 | 53.97 / 95.36 / 52.55 | 54.35 / 96.74 / 93.48 | 64.19 / 84.69 / 92.97 | 53.67 / 94.19 / 75.93 |
| 9. LN-Entropy † | 50.78 / 96.68 / 49.02 | 55.42 / 92.68 / 88.79 | 45.79 / 93.04 / 75.80 | 59.65 / 90.50 / 88.61 | 58.71 / 93.19 / 54.34 | 52.06 / 97.12 / 94.89 | 75.24 / 77.74 / 97.15 | 56.80 / 90.98 / 83.27 |
| 10. EigenScore †* | 54.05 / 95.03 / 49.80 | 56.32 / 91.35 / 90.19 | 48.71 / 95.03 / 79.66 | 63.59 / 90.29 / 92.79 | 60.32 / 90.71 / 53.58 | 61.17 / 90.36 / 97.25 | 71.04 / 80.90 / 95.48 | 59.31 / 90.52 / 79.82 |
| CoE-R (Ours) * | 60.74 / 90.48 / 58.64 | 64.81 / 88.46 / 92.85 | 54.69 / 95.58 / 84.00 | 71.92 / 74.76 / 94.55 | 65.71 / 91.52 / 60.40 | 72.35 / 71.79 / 98.48 | 75.54 / 76.19 / 97.72 | 66.54 / 84.11 / 83.74 |
| CoE-C (Ours) * | 55.49 / 92.20 / 53.15 | 58.47 / 89.42 / 90.70 | 55.11 / 91.71 / 84.43 | 70.87 / 80.58 / 94.32 | 66.70 / 87.68 / 61.45 | 73.32 / 69.58 / 98.43 | 74.88 / 76.19 / 97.59 | 64.98 / 83.90 / 82.87 |

et al., 2023), **Llama3-8B** (MetaAI, 2024), **Qwen1.5-7B** (Qwen-Team, 2024), **Qwen2-7B** (Yang et al., 2024), **Mistral-7B** (Jiang et al., 2023a). Additionally, to demonstrate the robustness of model parameter scaling, we also test two larger models with 70B+ parameters: **Llama3-70B** (MetaAI, 2024) and **Qwen2-72B** (Yang et al., 2024). Implementation details are shown in Appendix C.2.

**Baseline.** We select *ten* label-free self-evaluation baselines for fair comparisons: The first two represent typical paradigms that do not assess internal states: (1) **Verbal Confidence**, we select P(true) (Kadavath et al., 2022) for its versatility. (2) **Prompt-Sampling-Aggregation (PSA) Pipeline** (Xiong et al., 2024), we refer to Gao et al. (2024) to perturb input prompt with special tokens and aggregate sampling outputs based on *lexical similarity* (Lin et al., 2022c; Kuhn et al., 2023) with Rouge-L (Lin, 2004); The remaining eight are centered on the idea of uncertainty estimation, which assesses the internal state: (3) **Maximum Softmax Probability**; (4) **Perplexity** (Si et al., 2022); (5) **Entropy** (Huang et al., 2023); (6) **Temperature Scaling** (Shih et al., 2023); (7) **Energy** (Liu et al., 2020); (8) **Monte-Carlo Dropout** (Gal & Ghahramani, 2016); (9) **Length-normalized Entropy** (Malinin & Gales, 2020); (10) **Eigenscore** (Chen et al., 2024). Baseline details are shown in Appendix C.3. Among these, (3)-(7) require only single output distribution, (2) and (8)-(10) require multiple stochastic inferences for sampling, and (10) only utilize the hidden states.

**Evaluation.** We select **AUROC** (Boyd et al., 2013), **FPR95**, and **AUPR** (Manning, 1999) metrics to evaluate performances. AUROC focuses on the trade-off between TPR and FPR under different thresholds; FPR95 focuses on the rate of misclassifying samples when TPR reaches 95%; AUPR focuses on the trade-off between Precision and Recall, placing greater importance on the correct prediction of positive cases. These three metrics complement each other and can reflect the classification performances from different perspectives (Cen et al., 2021; Hendrycks et al., 2022). Additionally, we use the exact match to obtain the correctness labels 0/1 of LLM responses for evaluation.

## 4.2 MAIN RESULTS (TABLE 1)

**Method Comparisons.** First, our methods **achieve SOTA performances across almost all scenarios. In four domains, our CoE achieves an average improvement of 8.30%, 5.55%, and**

**5.52% across three metrics compared to the optimal baseline**. Notably, an AUROC value below 60 denotes a deviation from ideal performance (Xiong et al., 2024). **Our CoE exceeds this threshold in most scenarios, underscoring its practical value**. In contrast, most baselines fall short of this standard, particularly in the last three domains.

In contrast to the consistently optimal performance of our CoE method across nearly all scenarios, we observe that the performances of other baselines lack stability, with its effectiveness varying significantly across different scenarios. Below, we provide an analysis of other methods:

- First, we examine two typical paradigms (1-2) that do not assess internal states. Verbal confidence often exhibits poor performances, mainly due to a confirmed overconfidence issue (Zhang et al., 2024; Xiong et al., 2024). PSA pipeline lacks stability, it can either perform optimally or underperform. This may stem from unresolved issues related to effective consistency metrics (Manakul et al., 2023) and inadequate prompt robustness. See Appendix A.1.2 for more discussions.

- Secondly, we examine the uncertainty estimation methods (3-10) that assess internal states.

  - Single-output-based methods (3-7) possess the highest stability. While they do not always achieve best performances, they seldom rank the worst in any scenario. However, on the other hand, their performance limitations align with the assertion by Liu et al. (2024) that "the traditional uncertainty estimation can be extremely challenging when applied to LLMs due to the output diversity". Our chosen tasks include challenging datasets like MATH, where LLMs can generate solutions with thousands of tokens. This complexity greatly limits these methods.

  - In contrast, multiple-output-based methods (8-10) exhibit greater instability. For example, the LN-entropy method does not consistently surpass its base entropy version and can sometimes perform the worst, highlighting the inherent uncertainties associated with sampling.

  - We are especially interested in the performances of Eigenscore (10), as it also utilizes hidden state information. However, it falls short of ideal performance, particularly in the *Mathematical* domain. The motivation behind Eigenscore is the rich semantic information inherent in the embedding space can be utilized (Chen et al., 2024). However, Wang et al. (2024a) has found that the embedding modeling is often inaccurate for mathematical tasks, which may result in underperformance. In contrast, we emphasize the dynamic changes of hidden states, focusing on their behaviors within the latent space rather than on the specific state representations.

**Metric Consistency.** Among the three metrics, AUROC emphasizes the trade-off between TPR and FPR, FPR95 emphasizes the FPR, and AUPR emphasizes the prediction of positive cases. They fully simulate the various demands in real-life scenarios. In our experiments, we find that in most cases, our CoE method maintains the best performance across all three metrics simultaneously, except in a few cases where the trends of AUROC and FPR95 are inconsistent. This suggests that our method may focus more on positive cases, *i.e.*, correctly responded samples. Overall, **our CoE method can adequately adapt to various discrimination needs in real-world scenarios**.

**Domain Robustness.** In vertical comparisons between four domains, the average AUROC[1] performance improvements compared to the strongest baseline are **9.83%, 8.36%, 7.78%, 7.23%** from domains I to IV. Notably, **no domain shows a significantly lower improvement, highlighting our domain robustness**. More interestingly, the improvement in the *Mathematics* domain is noticeably greater, and it is exactly the most objective among the four domains, suggesting that our method is likely more effective in objective scenarios. This phenomenon aligns with human intuition: compared to solving subjective problems, humans' thinking path when solving objective problems tends to be less influenced by subjective feelings and biases. This reduces the subjective noise and enhances path systematic (Paul & Elder, 2019), thereby increasing the quantification precision of path features.

**Model-scaling Robustness.** In horizontal comparisons between 70B+ and 7B+ parameter models, we are surprised to find that even with a tenfold increase in parameters, our method still maintains its leading performance and even surpasses the 7B+ model performances in most cases. As the demand for large-scale LLMs surges in the industry, **the enhanced model scaling robustness allows our method for widespread deployment in real-world scenarios, ensuring its broad generalizability**.

---

[1] Note that the ratio of positive to negative examples varies across different scenarios, as it depends on the model's accuracy for each specific task. Therefore, when performing a vertical comparison between various domains of the same model or method, AUROC is the most appropriate metric for it is insensitive to the ratio of positive to negative examples compared to the other two metrics.

## 4.3 EXTENDED ANALYSIS

- *First, we further analyze the effectiveness of our CoE methods.*

**Component Ablation.** CoE scores consist of two components: magnitude and angle. To assess their impact on the combination metric, we conduct ablation studies. Table 2 presents the AUROC results for four 7B+ parameter models. We observe that in 14 out of 16 settings, the combination metric outperforms the individual components, indicating a positive influence from both components. Furthermore, when anomalies arise — such as in *Mathematics* and *Knowledge* domains with Qwen1.5-7B — CoE-R is more affected by these anomalies, whereas CoE-C demonstrates greater robustness. As a result, CoE-C offers more stable performance for real-world applications.

Table 2: Component ablation study of our CoE metric, we report AUROC results in four 7B+ models.

| Components | | Combination | | Llama3-8B-Instruct | | | | Qwen2-7B-Instruct | | | |
| Magnitude | Angle | CoE-R | CoE-C | Mathematics | Reasoning | Knowledge | Understanding | Mathematics | Reasoning | Knowledge | Understanding |
|---|---|---|---|---|---|---|---|---|---|---|---|
| ✓ | ✓ | ✓ | | 72.54 | **63.12** | **64.20** | **64.81** | 75.75 | **66.68** | **63.14** | **71.92** |
| ✓ | ✓ | | ✓ | 73.08 | 55.85 | 62.45 | 58.47 | **76.68** | 62.70 | 61.85 | 70.87 |
| ✓ | | | | **74.50** | 55.57 | 61.79 | 56.88 | 75.69 | 61.95 | 61.05 | 70.98 |
| | ✓ | | | 71.69 | 62.65 | 63.91 | 64.27 | 68.06 | 64.50 | 61.87 | 69.52 |
| Components | | Combination | | Llama2-7B-Instruct | | | | Qwen1.5-7B-Instruct | | | |
| Magnitude | Angle | CoE-R | CoE-C | Mathematics | Reasoning | Knowledge | Understanding | Mathematics | Reasoning | Knowledge | Understanding |
| ✓ | ✓ | ✓ | | 64.23 | 55.51 | **62.76** | **60.74** | 38.44 | 58.19 | 49.48 | 54.69 |
| ✓ | ✓ | | ✓ | 63.63 | **59.00** | 59.07 | 55.49 | **77.22** | **67.67** | **62.11** | **55.11** |
| ✓ | | | | 50.89 | 57.86 | 53.97 | 46.55 | 74.85 | 67.31 | 62.05 | 54.96 |
| | ✓ | | | **65.94** | 53.97 | 51.15 | 59.09 | 28.90 | 54.83 | 44.00 | 54.71 |

**Task Difficulty Exploration.** Tasks within the same domain can vary in difficulty, likely affecting metric performances. In our setup, we select two datasets of different difficulty levels for *Mathematics* and *Reasoning* domains, respectively. In particular, GSM8K and CommonsenseQA are low-difficulty datasets; MATH and TheoremQA are high-difficulty datasets for they require at least college knowledge. Details are shown in Appendix C.1. Figure 5 shows the AUROC results with the Qwen2-7B-Instruct model, where CoE has a slight edge in low-difficulty tasks, but demonstrates a significant advantage in high-difficulty tasks, and outperforms other baselines by large points. This indicates that CoE is more discriminative on more difficult tasks, which may be because the thinking paths are more complex on difficult tasks, increasing the potential informational features for CoE.

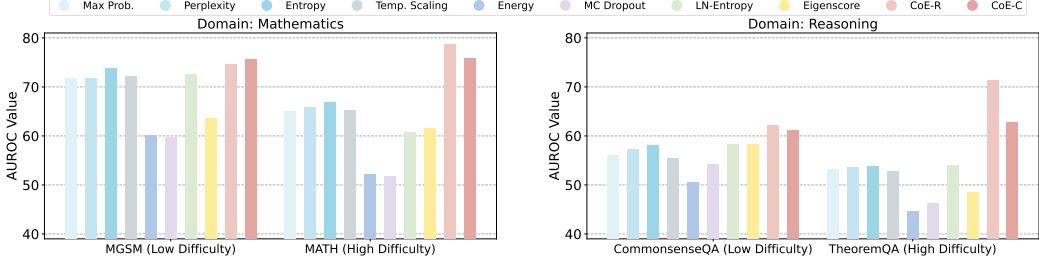

Figure 5: AUROC results (w/ Qwen2-7B-Instruct model) of all methods for varying difficulty tasks within the *Mathematics* and *Reasoning* domains. See Appendix D for other models.

- *Secondly, we analyze the reliability and stability of applying CoE methods in real-world scenarios.*

**Data Ratio Robustness.** Self-evaluation differs from other classification tasks because the ratio of positive to negative samples in each scenario is not balanced, it entirely depends on the accuracy $a$ of LLM responses. If we denote the number of positive samples in one dataset as $s_+$ and negative samples as $s_-$, then the ratio can be expressed as $s_+ : s_- = a$. To assess the performance robustness of CoE under different data ratios, we match all the CoE results in Table 1 with the response accuracy of the corresponding models on the corresponding datasets, then observe the AUROC results under different data ratios. Figure 6 shows the results, there is no significant performance drop in any particular area, especially in areas where $a < 0.2$ and $a > 0.8$, indicating that data imbalance does not adversely affect performances. This suggests that CoE is robust against varying data ratios.

**High Deployment Efficiency.** Existing methods have significant efficiency bottlenecks. Excluding the base LLM inference: For sampling-based methods, LLMs must perform at least one additional inference, so the inference time is the lower bound of its execution cost. For sampling-free output-based methods, they almost all require the output probability distribution, so the $\mathrm{SoftMax}$ computation

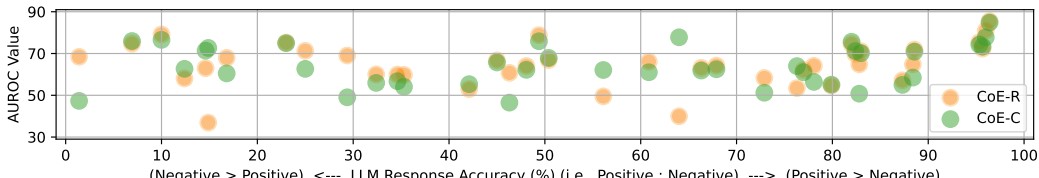

Figure 6: AUROC results under different LLM response accuracies (*i.e.*, data ratios).

is unavoidable, which involves large-scale exponential computations as the vocabulary size is huge (Mikolov, 2013). In practice, we find that when the task is difficult and the LLM response is long, this computation time will even be comparable to the inference time.

Table 3 compares the execution time (excluding the first base inference, which is mandatory) of our CoE method with the above two types of methods. Our method requires only simple addition, multiplication, and triangulation operations, **with execution costs at the millisecond level, and possesses prominent efficiency and stability advantages**.

Table 3: Execution Time(s) (w/ Llama3-8B-Instruct model) on GSM8K dataset.

| Base LLM Inference | 12.59 ±3.75 |
| SoftMax Computation | 10.32 ±3.51 |
| CoE Computation (ours) | 1.12e-03 ±5.64e-05 |

**Multilingual Scalability.** We also test the scalability of CoE in multilingual scenarios, which ensures that CoE is effective not only in English environments, thereby enhancing its universality. We select MGSM (Shi et al., 2022) dataset — a subset of the GSM8K with **11 language versions**. Figure 7 presents the results in the Llama3-8B-Instruct model, we find that **in almost all language scenarios, CoE demonstrates a performance advantage compared to baselines**. Notably, CoE shows a good performance in some low-resource languages (*e.g.*, bn), reflecting its adaptability to various language environments.

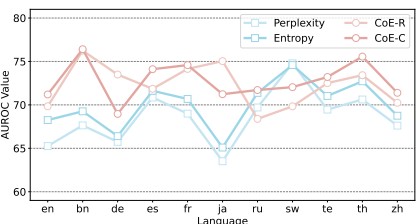

Figure 7: AUROC results (w/ Llama3-8B-Instruct model) on MGSM dataset with 11 language versions. Appendix D shows the results of other models.

## 5 THEORETICALLY REVISIT CoE-C AND CoE-R

**Monotonicity Analysis.** In Section 3, we pointed out that CoE-R lacks physical significance as a metric. However, its linearity effectively captures the monotonicity of both magnitude and angle features. In contrast, the monotonicity of these two features of CoE-R is not as apparent. We assume $n$ feature points, each represented as a pair of magnitude and angle $(L_i, \alpha_i)$ for $1 \leq i \leq n$, with $\boldsymbol{L} = [L_i]_{i=1}^n$ and $\boldsymbol{\alpha} = [\alpha_i]_{i=1}^n$. The final CoE feature is denoted as $\mathcal{F}(\boldsymbol{L}, \boldsymbol{\alpha})$. We consider the increments $\Delta L$ applied to $L_i$ and $\Delta\alpha$ applied to $\alpha_i$, with the CoE feature increments being

$$\Delta\mathcal{F}(L_i) = \mathcal{F}(L_i + \Delta L, \boldsymbol{L}_{-i}, \boldsymbol{\alpha}) - \mathcal{F}(L_i, \boldsymbol{L}_{-i}, \boldsymbol{\alpha}),$$
$$\Delta\mathcal{F}(\alpha_i) = \mathcal{F}(\alpha_i + \Delta\alpha, \boldsymbol{\alpha}_{-i}, \boldsymbol{L}) - \mathcal{F}(\alpha_i, +\Delta\alpha, \boldsymbol{\alpha}_{-i}, \boldsymbol{L}) \quad (8)$$

For CoE-R, its final feature $\mathcal{F}_R = \sum_{j=1}^n \frac{L_j}{n} + \sum_{j=1}^n \frac{\alpha_j}{n}$ ensures $\Delta\mathcal{F}_R(L_i) = \frac{\Delta L}{n} > 0$ and $\Delta\mathcal{F}_R(\alpha_i) = -\frac{\Delta\alpha}{n} < 0$. In contrast, The situation on CoE-C is relatively complex, we formalize its final feature $\mathcal{F}_C$ and computer its feature increments $\Delta\mathcal{F}_C(L_i)$ and $\Delta\mathcal{F}_C(\alpha_i)$ by following Eq.8 as below:

$$\mathcal{F}_C(\boldsymbol{L}, \boldsymbol{\alpha}) = \sqrt{\left(\frac{\sum_{j=1}^n L_j \cos\alpha_j}{n}\right)^2 + \left(\frac{\sum_{j=1}^n L_j \sin\alpha_j}{n}\right)^2} = \frac{1}{n}\sqrt{\sum_j L_j^2 + \sum_{k,t,k \neq t} 2L_k L_t \cos(\alpha_k - \alpha_t)}. \quad (9)$$

$$\Delta\mathcal{F}_C(L_i) = \frac{\Delta L \left(2L_i + \Delta L + \sum_{j, j \neq i} 2L_j \cos(\alpha_i - \alpha_j)\right)}{n^2 \mathcal{F}_C(L_i + \Delta L, \boldsymbol{L}_{-i}, \boldsymbol{\alpha}) + n^2 \mathcal{F}_C(\boldsymbol{L}, \boldsymbol{\alpha})}, \quad \Delta\mathcal{F}_C(\alpha_i) = \frac{-\sum_{i,j,i \neq j} L_i L_j \sin\left(\alpha_i - \alpha_j + \frac{\Delta\alpha}{2}\right) \sin(\Delta\alpha)}{n^2 \mathcal{F}_C(\alpha_i + \Delta\alpha, \boldsymbol{\alpha}_{-i}, \boldsymbol{L}) + n^2 \mathcal{F}_C(\boldsymbol{L}, \boldsymbol{\alpha})}. \quad (10)$$

See Appendix B.2.1 for the derivation of Eq.10. In practical inference, we statistically find that more than 98% of the cases fall within the range of $\alpha_i$ between 0 and $\pi/2$, this can also be intuitive in Figure 3. Consequently, it is nearly always true that $\Delta\mathcal{F}_C(L_i) > 0$. Additionally, the angle difference between correct and incorrect trajectories is often greater than the angle difference within a single trajectory, namely $|\alpha_i - \alpha_j|$. When $\Delta\alpha$ causes $\alpha_i$ to deviate from the current class feature, it tends to be sizable. As a result, $\sin\left(\alpha_i - \alpha_j + \frac{\Delta\alpha}{2}\right)$ will be greater than 0, leading to $\Delta\mathcal{F}_C(\alpha_i) < 0$.

Therefore, in practical scenarios, the CoE-C monotonicity of both magnitude and angle features is consistent with CoE-R and satisfies conclusions drawn in Section 2.2.

**Why CoE-C is More Robust Than CoE-R?** In Section 3, we pointed out that CoE-C may be more sensitive to outliers. This claim has been verified in the ablation study presented in Section 4.3. Here, we delve into the fundamental reasons behind the metric robustness from a theoretical perspective.

We already know that the magnitude changes of the CoE trajectory of a correct sample is more significant, which means that for an incorrect sample, if one $L_i$ of a feature point appears abnormally large, it will be easily misclassified as a correct sample. Therefore, if one CoE feature can better control the increment when facing this situation, it will reduce the risk of misclassification. Formally, we compare $\Delta\mathcal{F}_R(L_i)$ and $\Delta\mathcal{F}_C(L_i)$, the smaller one CoE metric possesses stronger robustness. We first deflate the lower bound of $\Delta\mathcal{F}_C(L_i)$ by fixing the principal element $L_i$:

$$\mathcal{F}_C(\boldsymbol{L}, \boldsymbol{\alpha}) = \frac{1}{n}\sqrt{\sum_j L_j^2 + \sum_{k,t,k \neq t} 2L_k L_t \cos(\alpha_k - \alpha_t)} \geq \frac{1}{n}\left(L_i + \sum_{j,j \neq i} L_j \cos(\alpha_i - \alpha_j)\right) \quad (11)$$

Then, we use this deflation bound to further deflate the $\Delta\mathcal{F}_C(L_i)$ of Eq.8:

$$\Delta\mathcal{F}_C(L_i) \leq \frac{\Delta L\left(2L_i + \Delta L + \sum_{j,j \neq i} 2L_j \cos(\alpha_i - \alpha_j)\right)}{n^2 \cdot \frac{1}{n}\left(L_i + \Delta L + \sum_{j,j \neq i} L_j \cos(\alpha_i - \alpha_j)\right) + n^2 \cdot \frac{1}{n}\left(L_i + \sum_{j,j \neq i} L_j \cos(\alpha_i - \alpha_j)\right)} = \frac{\Delta L}{n} \quad (12)$$

We find that the right side of Eq.12 is exactly $\mathcal{F}_R(L_i)$, which implies $\mathcal{F}_C(L_i) \leq \mathcal{F}_R(L_i)$, proving that CoE-C is more robust than CoE-R. The complete derivation can be found in Appendix B.2.2.

## 6 RELATED WORK AND DISCUSSION

Our research focuses on label-free self-evaluation, where the uncertainty estimation in deep neural networks (Gal & Ghahramani, 2016; Guo et al., 2017) and their variants in the era of LLMs (Huang et al., 2023; Kuhn et al., 2023) are closely related to us, we categorize them as white-box methods. Additionally, the two typical paradigms that do not access internal states are classified as black-box methods (Manakul et al., 2023; Li et al., 2024c). Among these, white-box and black-box research tracks are usually orthogonal to each other (Li et al., 2024c). They all emphasize the "label-free" condition, and we present a detailed discussion about these related works in Appendix A.1.

From the perspective of research ideas, our research involves the usage of hidden state information. Many existing studies typically utilize hidden states to train correctness-label-based probing classifiers. They can learn useful hidden state features of specific datasets or error types, but their generalization ability on out-of-distribution (OOD) data is unpredictable. Despite the differing research intents, considering the overlap of research ideas, we also conduct simple comparisons with them.

We select two recent works: (1) ITI (Li et al., 2024a), which is trained on the TruthfulQA dataset (Lin et al., 2022b); and (2) MIND (Su et al., 2024), which is trained with hallucinated data sourced from Wikipedia. Both methods focus on detecting factual errors, making mathematics and reasoning tasks likely OOD for them. We select four datasets: (1) TruthfulQA (Lin et al., 2022b) as it is in-distribution (ID) for both ITI and MIND and even fitted by ITI; and (2-4) GSM8k, MATH, and TheoremQA (used in our main experiments), they are OOD for both ITI and MIND.

Table 4 presents the AUROC results with the Llama3-8B-Instruct model. On the TruthfulQA dataset, ITI outperforms CoE, which is expected given that it is well-suited to its correctness features. However, in the other three datasets, ITI and MIND exhibit significant declines with more than 20 points of MIND and 30 points of ITI due to being OOD relative to their training data. In contrast, our CoE is not tailored to specific dataset features and maintains consistent applicability across diverse datasets, making it robust for complex real-world scenarios. We discuss more related work in this research area in Appendix A.2.

Table 4: AUROC Results of our label-free CoE and label-based ITI and MIND (w/ Llama3-8B-Instruct model) on four datasets.

|  | *TruthfulQA* | GSM8K | MATH | TheoremQA |
|---|---|---|---|---|
| ITI | **83.48** | 47.49 | 46.02 | 48.35 |
| MIND | 74.52 | 51.28 | 50.67 | 43.96 |
| CoE-R (ours) | 72.21 | 69.84 | **75.23** | **67.94** |
| CoE-C (ours) | 74.74 | **71.20** | 74.95 | 60.47 |

## 7 CONCLUSION

In summary, we propose a lightweight self-evaluation method for LLMs. It does not access the output text or probability distribution but utilizes the progressive chain of all hidden states in the latent space instead, which we term Chain-of-Embedding (CoE). Our method exhibits strong performance and robustness across various models, domains, task difficulties, and languages. Its low computational cost also ensures real-time deployments for large-scale feedback needs in practical scenarios.

## LIMITATION

Our method cannot be extended to a black-box version because of the need to access the internal hidden states, this means that our method cannot be applied to closed-source models such as GPT-4 (Achiam et al., 2023) for the time being. However, from the perspective of research senses, white-box approaches are more helpful in interpreting the response mechanisms within LLMs, while black-box approaches are more intuitive and better suited to cope with closed-source protocols. Therefore, we advocate recognizing the parallel contributions of the two research tracks and flexibly choosing between the two types of methods for use in real-world scenarios.

## ETHNICS STATEMENT

The data and models used in this work are sourced from the official version of the original paper, and we strictly adhere to the provided usage protocol. Regarding the data, no modifications have been made to the original dataset, so they would not involve any sensitive content. From the perspective of research intent, our research aims to detect the incorrect responses generated by LLM in real-world deployment, which has promising implications for social safety.

## REPRODUCIBILITY

All models employed in the experiments are official checkpoints, and all implementation details including model sources, hyperparameters, hardware requirements, and prompt instructions are presented in Appendix C.2 to ensure the reproducibility of our method.

## ACKNOWLEDGEMENT

This work was supported by the Alibaba Research Intern Program and Alibaba Innovative Research Program. This work was supported by the General Program of National Natural Science Foundation of China (62176153). This work was also supported in part by the Science and Technology Development Fund of Macau SAR (Grant Nos. 0007/2024/AKP, FDCT/0070/2022/AMJ, FDCT/060/2022/AFJ), and the UM and UMDF (Grant Nos. MYRG-GRG2023-00006-FST-UMDF, MYRG-GRG2024-00165-FST-UMDF, EF2024-00185-FST, EF2023-00151-FST, EF2023-00090-FST).

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

## A  ADDITIONAL RELATED WORK AND DISCUSSION

Our research topic is about LLM self-evaluation, where the **"self"** hides two key constraints: (1) this evaluation is **label-free** — suppose there is an urgent need for evaluation, only a large number of crawled questions, and there is no time and manpower to label high-quality answers. online feedback generated when the model answers these questions to determine the correctness of its answers; (2) this evaluation is not allowed to have any external tools involved, in other words, **no external scorers or trainers**. This is a necessary guarantee for the industry to save the cost of model deployment as well as to ensure its real-time and scalability when facing large-scale evaluation needs.

### A.1  LABEL-FREE LLM SELF-EVALUATION

#### A.1.1  WHITE-BOX METHODS

The precursor to LLM white-box evaluation can be traced back to *Uncertainty Estimation (UE)* (Lakshminarayanan et al., 2017) during the era of deep neural networks. At that time, the concept of uncertainty was commonly understood as output probabilities (Gal & Ghahramani, 2016; Guo et al., 2017; Desai & Durrett, 2020), which could be used to measure the model's confidence in generating that output. These methods all encounter a key issue that their model output may be overconfident, this is because the KL-divergence optimization adopted by model training (including language models) forces models to assign non-zero (often quite high) probabilities to all training samples (Minka et al., 2005; Malinin & Gales, 2019). To cover the low-probability regions in the data distribution, LLMs adopt the zero-avoiding solution (Deasy et al., 2020) and systematically overestimate the probabilities of almost all text sequences even if they are ill-formed (Ji et al., 2023). Therefore, confidence calibration is always a popular research topic, aimed at adjusting the output logits or probabilities to align with the true confidence levels.

The emergence of LLMs has introduced new challenges of uncertainty estimation in natural language generation tasks (Shorinwa et al., 2024) due to their diverse outputs (Liu et al., 2024). Meanwhile, rich evaluation requirements transform this estimation from a continuous probabilistic target (confidence) into a discrete binary target (correctness), emphasizing the importance of accurately predicting the correctness of LLMs' output (Manakul et al., 2023; Li et al., 2024c). Research in this area remains sparse, primarily adhering to conventional ideas of uncertainty estimation that leverage the output logits or probability distributions produced by LLMs. Notably, Si et al. (2022) pioneered the use of perplexity metrics with GPT-3; Huang et al. (2023) systematically assessed how entropy can evaluate self-evaluation in language models; (Plaut et al., 2024) used softmax probabilities to predict the correctness of multiple-choice questions; Hendrycks et al. (2022) explored temperature scaling for output probabilities. Apart from the challenges posed by output diversity, a more crucial distinction between language models and traditional deep networks is the rich semantic information inherent in LLM outputs (Liu et al., 2024). To address this, Kuhn et al. (2023) and Farquhar et al. (2024) investigated the concept of semantic entropy, and Duan et al. (2023) also introduced a shift towards utilizing semantic information in entropy calculations. These works primarily focus on a specific type of factual error, namely hallucinations.

Beyond traditional uncertainty estimation ideas, LLMs contain rich hidden state information that can enhance self-evaluation. However, existing research typically involves training external classifiers (Discussed in Section A.2), and there has been limited exploration of label-free self-evaluation methods utilizing hidden states. One such study by (Chen et al., 2024) measures the covariance of hidden states at a certain layer across multiple samples based on the idea of internal consistency (Liang et al., 2024). Our research fills a significant gap in this research area by going beyond sampling methods and comprehensively examining how the trajectory of changes in model hidden states can inform LLM self-evaluation.

#### A.1.2  BLACK-BOX METHODS

LLM black-box self-evaluation methods primarily fall into two paradigms: *Verbal Confidence (VC)* and *Prompt-Sampling-Aggregation (PSA)*.

*VC* is the more straightforward approach, leveraging the instruction-following capabilities of LLMs, enhanced by RLHF (Ouyang et al., 2022), to generate confidence scores through well-crafted prompts.

This paradigm includes various general methodologies that directly ask LLMs for their confidence using multi-stage pipelines (Lin et al., 2022a; Manakul et al., 2023; Tian et al., 2023; Li et al., 2024b; Wang et al., 2024b; Taghanaki & Monteiro, 2025) with some additional technical tools like reflection mechanisms (Feng et al., 2024). Besides this, there are domain-specific strategies tailored for areas such as code generation (Zhou et al., 2023) and fact-checking (Lin et al., 2023). Another notable indirect verbal method is the P(True) (Kadavath et al., 2022), which assesses the likelihood that the next token output is True. *PSA* pipeline (Xiong et al., 2024) estimates confidence by perturbing prompts (Jiang et al., 2023b; Gao et al., 2024; Liu et al., 2025) or by stochastic decoding (Si et al., 2022; Wang et al., 2023) to generate multiple outputs. It then assesses the consistency of these outputs.

The advantages of these methods are their intuitive principles and the lack of constraints from closed-source licenses. However, they also face some unresolved issues. First, it has been confirmed that *VC* suffers from overconfidence — LLMs tend to assign high scores to their own outputs (Zhang et al., 2024; Xiong et al., 2024), which implicitly decreases the credibility of this approach. On the other hand, the *PSA* pipeline is considered incapable of finding an effective consistency measurement (Manakul et al., 2023; Zhang et al., 2023), which will increase the instability of this method when deployed in diverse scenarios.

In our view, the issue of *PSA* reflects the essential flaw of poor robustness in the selection of sub-modules: The key components of *PSA* lie in prompt sampling and multiple-answer aggregation strategies. Prompt sampling is mainly achieved through rephrasing, but different research has given different rephrase prompt designs, and some tasks only focus on specific types of tasks, such as multiple-choice questions (Jiang et al., 2023b). This makes it uncertain how to design the most suitable rephrasing prompt when facing a new task; Multiple-answer aggregation techniques mainly assess the consistency between multiple answers, and the simplest way is to match precise answers and calculate frequencies directly. However, this only applies to deterministic answers, and for some generation tasks or answers involving the problem-solving process, the semantic similarity between answers is also crucial. This makes the best aggregation techniques uncertain.

To test the strategy combination robustness, we select two prompt perturbation strategies (rephrased by two people) and two aggregation strategies (exact match and semantic similarity) to test the consistency of self-evaluation results in four (2*2) scenarios. We use the Llama3-8B-Instruct and Qwen2-7B-Instruct model for testing in four domains and used AUROC as the metric. Figure 8 shows the ranking results, which reflect the AUROC ranking for the four strategies under the same domain. We find that no single strategy remains ahead in all settings, which means that applying *PSA* in diverse real-world scenarios is unpredictable.

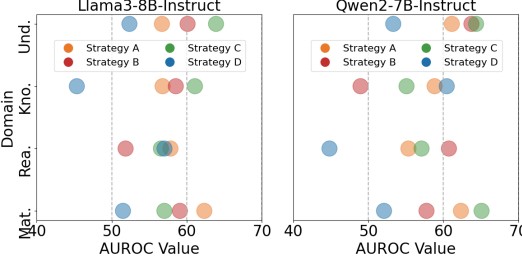

Figure 8: AUROC ranking of different prompt sampling and consistency aggregation strategies in four domains with two models. It is clear that no one combined strategy outperforms others in any scenario.

## A.2 Hidden States for Label-based Self-Evaluation

Setting aside the intended limitations of being "label-free", the rich information of hidden states is often utilized in supervised correctness estimation studies, which typically focus on a particularly important type of error — factual errors, commonly referred to as "hallucinations". These studies usually use certain hidden states as input and label correctness to train an external classifier. Notably, Mielke et al. (2022); Sky et al. (2024) trained a classifier to predict hallucinations; Li et al. (2024a) conducted probing experiments on hallucination issues using the specific TruthfulQA dataset (Lin et al., 2022b) and extracted potential detection information from attention modules for training; Su et al. (2024) collected hallucination corpora from Wikipedia for annotation and training. Some probe analyses also explained the differences in the hidden states exhibited by language models when producing correct and incorrect answers (Azaria & Mitchell, 2023; Liu et al., 2023; Duan et al., 2024), but they did not offer extended insights into the correctness estimation.

These methods leverage hidden states to provide valuable correctness estimation insights. However, a common limitation is that they all involve training on supervised corpora. This goes against the intention of being "label-free". More importantly, classifier training is domain-restricted: It is often task-specific (trained on particular datasets) or type-specific (trained on factual errors), which results in unstable performance when faced with Out-of-Distribution (OOD) data, hindering their scalability. In Section 6, we have demonstrated this argument.

## B  MORE ANALYSIS ABOUT CoE SCORES

### B.1  ALGORITHMIC PROCESS OF CoE SCORES

---

**Algorithm 1** CoE-R Computation

---

**Input:**  $L$: The number of hidden layers
  $d$: Embedding dimension
  $\boldsymbol{h}_0, \boldsymbol{h}_1, \cdots, \boldsymbol{h}_L \in \mathbb{R}^d$: All hidden states
1:  $\mathcal{Z}_{\text{Mag}} \leftarrow ||\boldsymbol{h}_L - \boldsymbol{h}_0||_2$
2:  $\mathcal{Z}_{\text{Ang}} \leftarrow \arccos\left(\frac{\boldsymbol{h}_0^\top \boldsymbol{h}_L}{||\boldsymbol{h}_0||_2 \cdot ||\boldsymbol{h}_L||_2}\right)$
3:  CoE-R $\leftarrow 0$
4:  **for** $l \leftarrow 0$ **to** $L-1$ **do**
5:    Mag $\leftarrow ||\boldsymbol{h}_{l+1} - \boldsymbol{h}_l||_2$
6:    Ang $\leftarrow \arccos\left(\frac{\boldsymbol{h}_{l+1}^\top \boldsymbol{h}_l}{||\boldsymbol{h}_{l+1}||_2 \cdot ||\boldsymbol{h}_l||_2}\right)$
7:    CoE-R $\leftarrow$ CoE-R $+ \frac{\text{Mag}}{\mathcal{Z}_{\text{Mag}}} - \frac{\text{Ang}}{\mathcal{Z}_{\text{Ang}}}$
8:  **end for**
9:  CoE-R $\leftarrow \frac{\text{CoE-R}}{L}$
**Output:**  CoE-R

---

---

**Algorithm 2** CoE-C Computation

---

**Input:**  $L$: The number of hidden layers
  $d$: Embedding dimension
  $\boldsymbol{h}_0, \boldsymbol{h}_1, \cdots, \boldsymbol{h}_L \in \mathbb{R}^d$: All hidden states
1:  $\mathcal{Z}_{\text{Mag}} \leftarrow ||\boldsymbol{h}_L - \boldsymbol{h}_0||_2$
2:  CoE-C $\leftarrow 0$
3:  AvgRe, AvgIm $\leftarrow 0, 0$
4:  **for** $l \leftarrow 0$ **to** $L-1$ **do**
5:    Mag $\leftarrow ||\boldsymbol{h}_{l+1} - \boldsymbol{h}_l||_2$
6:    Ang $\leftarrow \arccos\left(\frac{\boldsymbol{h}_{l+1}^\top \boldsymbol{h}_l}{||\boldsymbol{h}_{l+1}||_2 \cdot ||\boldsymbol{h}_l||_2}\right)$
7:    Re $\leftarrow \frac{\text{Mag}}{\mathcal{Z}_{\text{Mag}}} \cos(\text{Ang})$
8:    Im $\leftarrow \frac{\text{Mag}}{\mathcal{Z}_{\text{Mag}}} \sin(\text{Ang})$
9:    AvgRe $\leftarrow$ AvgRe $+$ Re
10:   AvgIm $\leftarrow$ AvgIm $+$ Im
11: **end for**
12: CoE-C $\leftarrow \sqrt{\left(\frac{\text{AvgRe}}{L}\right)^2 + \left(\frac{\text{AvgIm}}{L}\right)^2}$
**Output:**  CoE-C

---

## B.2 Theoretical Analysis of CoE-C and CoE-R

### B.2.1 Derivation of CoE Features and Increments

We assume $n$ feature points, each represented as a pair of magnitude and angle $(L_i, \alpha_i)$ for $1 \le i \le n$, with $\boldsymbol{L} = [L_i]_{i=1}^n$ and $\boldsymbol{\alpha} = [\alpha_i]_{i=1}^n$. The final CoE feature is denoted as $\mathcal{F}(\boldsymbol{L}, \boldsymbol{\alpha})$. We consider the increments $\Delta L$ applied to $L_i$ and $\Delta \alpha$ applied to $\alpha_i$, with the CoE feature increments being

$$
\begin{aligned}
\Delta \mathcal{F}(L_i) &= \mathcal{F}(L_i + \Delta L, \boldsymbol{L}_{-i}, \boldsymbol{\alpha}) - \mathcal{F}(L_i, \boldsymbol{L}_{-i}, \boldsymbol{\alpha}), \\
\Delta \mathcal{F}(\alpha_i) &= \mathcal{F}(\alpha_i + \Delta \alpha, \boldsymbol{\alpha}_{-i}, \boldsymbol{L}) - \mathcal{F}(\alpha_i, +\Delta \alpha, \boldsymbol{\alpha}_{-i}, \boldsymbol{L}).
\end{aligned}
\tag{13}
$$

For CoE-R, its final feature $\mathcal{F}_R(\boldsymbol{L}, \boldsymbol{\alpha})$ and feature increments $\Delta \mathcal{F}_R(L_i), \Delta \mathcal{F}_R(\alpha_i)$ are:

$$
\mathcal{F}_R(\boldsymbol{L}, \boldsymbol{\alpha}) = \sum_{j=1}^n \frac{L_j}{n} + \sum_{j=1}^n \frac{\alpha_j}{n},
\tag{14}
$$

$$
\Delta \mathcal{F}_R(L_i) = \frac{\Delta L}{n},
\tag{15}
$$

$$
\Delta \mathcal{F}_R(\alpha_i) = -\frac{\Delta \alpha}{n}.
\tag{16}
$$

For CoE-C, its final feature $\mathcal{F}_R(\boldsymbol{L}, \boldsymbol{\alpha})$ and feature increments $\Delta \mathcal{F}_C(L_i), \Delta \mathcal{F}_C(\alpha_i)$ are:

$$
\mathcal{F}_C(\boldsymbol{L}, \boldsymbol{\alpha}) = \sqrt{\left( \frac{\sum_{j=1}^n L_j \cos \alpha_j}{n} \right)^2 + \left( \frac{\sum_{j=1}^n L_j \sin \alpha_j}{n} \right)^2} = \frac{1}{n} \sqrt{\sum_j L_j^2 + \sum_{k,t,k\ne t} 2 L_k L_t \cos(\alpha_k - \alpha_t)},
\tag{17}
$$

$$
\begin{aligned}
&\Delta \mathcal{F}_C(L_i) \\
&= \frac{1}{n} \sqrt{\left( \sum_{j=1,j\ne i}^n L_j^2 \right) + (L_i + \Delta L)^2 + \left( \sum_{k,t,k\ne t\ne i} 2 L_k L_t \cos(\alpha_k - \alpha_t) \right) + \sum_{j,j\ne i} 2(L_i + \Delta L) L_j \cos(\alpha_i - \alpha_j)} - \frac{1}{n} \sqrt{\sum_j L_j^2 + \sum_{k,t,k\ne t} 2 L_k L_t \cos(\alpha_k - \alpha_t)} \\
&= \frac{1}{n} \frac{(L_i + \Delta L)^2 - L_i^2 + \sum_{j,j\ne i} 2\Delta L L_j \cos(\alpha_i - \alpha_j)}{\sqrt{\left( \sum_{j=1,j\ne i}^n L_j^2 \right) + (L_i + \Delta L)^2 + \left( \sum_{k,t,k\ne t\ne i} 2 L_k L_t \cos(\alpha_k - \alpha_t) \right) + \sum_{j,j\ne i} 2(L_i + \Delta L) L_j \cos(\alpha_i - \alpha_j)} + \sqrt{\sum_j L_j^2 + \sum_{k,t,k\ne t} 2 L_k L_t \cos(\alpha_k - \alpha_t)}} \\
&= \frac{\Delta L \left( 2 L_i + \Delta L + \sum_{j,j\ne i} 2 L_j \cos(\alpha_i - \alpha_j) \right)}{n^2 \mathcal{F}(L_i + \Delta L) + n^2 \mathcal{F}(L_i)},
\end{aligned}
\tag{18}
$$

$$
\begin{aligned}
&\Delta \mathcal{F}_C(\alpha_i) \\
&= \frac{1}{n} \sqrt{\sum_j L_j^2 + \left( \sum_{k,t,k\ne t\ne i} 2 L_k L_t \cos(\alpha_k - \alpha_t) + \sum_{j,j\ne i} 2 L_i L_j \cos(\alpha_i + \Delta \alpha - \alpha_j) \right)} - \frac{1}{n} \sqrt{\sum_j L_j^2 + \sum_{k,t,k\ne t} 2 L_k L_t \cos(\alpha_k - \alpha_t)} \\
&= \frac{1}{n} \frac{\sum_{j,j\ne i} 2 L_i L_j \cos(\alpha_i + \Delta \alpha - \alpha_j) - \sum_{j,j\ne i} 2 L_i L_j \cos(\alpha_i - \alpha_j)}{\sqrt{\sum_j L_j^2 + \left( \sum_{k,t,k\ne t\ne i} 2 L_k L_t \cos(\alpha_k - \alpha_t) + \sum_{j,j\ne i} 2 L_i L_j \cos(\alpha_i + \Delta \alpha - \alpha_j) \right)} + \sqrt{\sum_j L_j^2 + \sum_{k,t,k\ne t} 2 L_k L_t \cos(\alpha_k - \alpha_t)}} \\
&= \frac{\sum_{i,j,i\ne j} 2 L_i L_j (\cos(\alpha_i + \Delta \alpha - \alpha_j) - \cos(\alpha_i - \alpha_j))}{n^2 \mathcal{F}(\alpha_i + \Delta \alpha) + n^2 \mathcal{F}(\alpha_i)} \\
&= \frac{-\sum_{i,j,i\ne j} L_i L_j \sin\left( \alpha_i - \alpha_j + \frac{\Delta \alpha}{2} \right) \sin(\Delta \alpha)}{n^2 \mathcal{F}(\alpha_i + \Delta \alpha) + n^2 \mathcal{F}(\alpha_i)}.
\end{aligned}
\tag{19}
$$

### B.2.2 Robustness Analysis

We already know that the magnitude changes of the CoE trajectory of a correct sample is more significant, which means that for an incorrect sample, if one $L_i$ of a feature point appears abnormally large, it will be easily misclassified as a correct sample. Therefore, if one CoE feature can better control the increment when facing this situation, it will reduce the risk of misclassification. Formally, we compare $\Delta \mathcal{F}_R(L_i)$ and $\Delta \mathcal{F}_C(L_i)$, the smaller one CoE metric possesses stronger robustness.

We find that $\Delta \mathcal{F}_C(L_i)$, as defined in Eq.18, includes $\mathcal{F}_C(\boldsymbol{L}, \boldsymbol{\alpha})$ (Eq.17). Therefore, before deflating $\Delta \mathcal{F}_C(L_i)$, we can first deflate $\mathcal{F}_C(\boldsymbol{L}, \boldsymbol{\alpha})$ and obtain the lower bound as follows:

$$\mathcal{F}_C(\boldsymbol{L}, \boldsymbol{\alpha})$$

$$= \frac{1}{n}\sqrt{L_i^2 + \sum_{j=1,j\neq i}^{n} L_j^2 + L_i\sum_{j,j\neq i} 2L_j\cos(\alpha_i-\alpha_j) + \sum_{k,t,k\neq t\neq i} 2L_k L_t\cos(\alpha_k-\alpha_t)}$$

$$= \frac{1}{n}\sqrt{\left(L_i + \sum_{j,j\neq i} L_j\cos(\alpha_i-\alpha_j)\right)^2 + \sum_{j=1,j\neq i}^{n} L_j^2 + \sum_{k,t,k\neq t\neq i} 2L_k L_t\cos(\alpha_k-\alpha_t) - \left(\sum_{j,j\neq i} L_j\cos(\alpha_i-\alpha_j)\right)^2}$$

$$= \frac{1}{n}\sqrt{\left(L_i + \sum_{j,j\neq i} L_j\cos(\alpha_i-\alpha_j)\right)^2 + \sum_{j=1,j\neq i}^{n} L_j^2 + \sum_{k,t,k\neq t\neq i} 2L_k L_t\cos(\alpha_k-\alpha_t) - \left(\sum_{j=1,j\neq i} L_j^2\cos^2(\alpha_i-\alpha_j) + \sum_{k,t,k\neq t\neq i} 2L_k L_t\cos(\alpha_i-\alpha_k)\cos(\alpha_i-\alpha_t)\right)}$$

$$= \frac{1}{n}\sqrt{\left(L_i + \sum_{j,j\neq i} L_j\cos(\alpha_i-\alpha_j)\right)^2 + \sum_{j=1,j\neq i} L_j^2[1-\cos^2(\alpha_i-\alpha_j)] + \sum_{k,t,k\neq t\neq i} 2L_k L_t\left[\cos(\alpha_k-\alpha_t) - \cos(\alpha_i-\alpha_k)\cos(\alpha_i-\alpha_t)\right]}$$

$$= \frac{1}{n}\sqrt{\left(L_i + \sum_{j,j\neq i} L_j\cos(\alpha_i-\alpha_j)\right)^2 + \sum_{j=1,j\neq i} L_j^2\sin^2(\alpha_i-\alpha_j) + \sum_{k,t,k\neq t\neq i} 2L_k L_t\sin(\alpha_i-\alpha_k)\sin(\alpha_i-\alpha_t)}$$

$$= \frac{1}{n}\sqrt{\left(L_i + \sum_{j,j\neq i} L_j\cos(\alpha_i-\alpha_j)\right)^2 + \left(\sum_{j=1,j\neq i} L_j\sin(\alpha_i-\alpha_j)\right)^2}$$

$$\geq \frac{1}{n}\left(L_i + \sum_{j,j\neq i} L_j\cos(\alpha_i-\alpha_j)\right).$$

When $\alpha_i = \alpha_j$ for all $1 \leq j \leq n, j \neq i$, $\mathcal{F}_C(\boldsymbol{L}, \boldsymbol{\alpha})$ achieves its lower bound. We can then use this lower bound to deflate the $\Delta\mathcal{F}(L_i)$ as follows:

$$
\begin{aligned}
\Delta\mathcal{F}_C(L_i) &= \frac{\Delta L \left(2L_i + \Delta L + \sum_{j,j \neq i} 2L_j \cos(\alpha_i - \alpha_j)\right)}{n^2 \mathcal{F}(L_i + \Delta L) + n^2 \mathcal{F}(L_i)} \\
&\leq \frac{\Delta L \left(2L_i + \Delta L + \sum_{j,j \neq i} 2L_j \cos(\alpha_i - \alpha_j)\right)}{n^2 \cdot \frac{1}{n}\left(L_i + \Delta L + \sum_{j,j \neq i} L_j \cos(\alpha_i - \alpha_j)\right) + n^2 \cdot \frac{1}{n}\left(L_i + \sum_{j,j \neq i} L_j \cos(\alpha_i - \alpha_j)\right)} \\
&= \frac{\Delta L \left(2L_i + \Delta L + \sum_{j,j \neq i} 2L_j \cos(\alpha_i - \alpha_j)\right)}{n\left(2L_i + \Delta L + \sum_{j,j \neq i} 2L_j \cos(\alpha_i - \alpha_j)\right)} \\
&= \frac{\Delta L}{n}.
\end{aligned}
\tag{20}
$$

We can derive that $\Delta\mathcal{F}_C(L_i) \leq \frac{\Delta L}{n}$. Luckily, $\frac{\Delta L}{n}$ is just the $\Delta\mathcal{F}_R(L_i)$ as defined in Eq.15. Thus $\Delta\mathcal{F}_C(L_i) \leq \Delta\mathcal{F}_R(L_i)$ is proved, *i.e.*, it is proved that CoE-C is more robust than CoE-R.

## C    EXPERIMENTAL SETUP DETAILS

### C.1    DATASET

We select six datasets across four domains in our main experiments. Table 5-10 present some dataset samples and corresponding LLM output (Produced by Llama3-8B-Instruct model) of each dataset. We describe each dataset's information as follows:

- *Mathematics Domain*

  - **GSM8K** (Cobbe et al., 2021) consists of linguistically diverse math word problems specifically designed for grade school students and created by human writers. It contains 1318 test problems, each requiring 2-8 steps to solve. The challenges primarily involve a sequence of basic arithmetic operations (addition, subtraction, multiplication, and division) to derive the final answer.

  - **MATH** (Hendrycks et al., 2021) consists of challenging competition mathematics problems. It contains 5000 test problems from diverse domains including algebra, geometry, counting and probability, number theory, and precalculus.

- *Reasoning Domain*

  - **CommonsenseQA** (Talmor et al., 2019) consists of diverse commonsense question answering questions. It contains 1221 test problems, each with 5 choices. This dataset was generated by Amazon Mechanical Turk workers.

  - **TheoremQA** (Chen et al., 2023) consists of various question-answering questions driven by STEM theorems. It contains 800 test problems and covers 350+ theorems spanning across Math, EE&CS, Physics and Finance. The dataset is collected by human experts with very high quality.

- *Knowledge Domain*

  - **MMLU** (Hendrycks et al., 2020) consists of diverse questions that measure knowledge acquired during pretraining. This dataset covers 57 subjects across STEM, the humanities, the social sciences, and more. It ranges in difficulty from an elementary level to an advanced professional level, can test both world knowledge and problem-solving ability.

- *Understanding Domain*

  - **Belebele** (Bandarkar et al., 2023) consists of multilingual, multitask, and multidisciplinary reading comprehension questions, with each language containing 900 article-question pairs. We use the English version of this dataset.

Table 5: Dataset case and LLM output from GSM8K dataset (*Mathematics* Domain)

| Question |
|---|
| Janet's ducks lay 16 eggs per day. She eats three for breakfast every morning and bakes muffins for her friends every day with four. She sells the remainder at the farmers' market daily for $2 per fresh duck egg. How much in dollars does she make every day at the farmers' market? |
| **True Answer** |
| 18 |
| **LLM Response (After following the instruction in Section C.2.3)** |
| Let's break down the problem step by step:
1. Janet's ducks lay 16 eggs per day.
2. She eats 3 eggs for breakfast, so she has 16 - 3 = 13 eggs left.
3. She bakes muffins with 4 eggs, leaving her with 13 - 4 = 9 eggs.
4. She sells the remaining 9 eggs at the farmers' market for $2 per egg.
5. To find out how much she makes, multiply the number of eggs sold (9) by the price per egg ($2): 9 x $2 = $18.
Answer: 18 |

Table 6: Dataset case and LLM output from MATH dataset (*Mathematics* Domain)

| Question |
| --- |
| The sets $A = \{z : z^{18} = 1\}$ and $B = \{w : w^{48} = 1\}$ are both sets of complex roots of unity. The set $C = \{zw : z \in A \text{ and } w \in B\}$ is also a set of complex roots of unity. How many distinct elements are in $C$? |
| **True Answer** |
| 144 |
| **LLM Response (After following the instruction in Section C.2.3)** |
| Let's break this down step by step! |
| 1. The set $A = \{z : z^{18} = 1\}$ contains all complex roots of unity that have a power of 18 equal to 1. This means that $A$ contains 18th roots of unity, which are the complex numbers that satisfy $z^{18} = 1$. |
| 2. The set $B = \{w : w^{48} = 1\}$ contains all complex roots of unity that have a power of 48 equal to 1. This means that $B$ contains 48th roots of unity, which are the complex numbers that satisfy $w^{48} = 1$. |
| 3. The set $C = \{zw : z \in A \text{ and } w \in B\}$ contains all complex numbers that are the product of a complex number from $A$ and a complex number from $B$. Since $A$ and $B$ are sets of complex roots of unity, the product of any two complex numbers from $A$ and $B$ will also be a complex root of unity. |
| 4. Now, let's think about the possible products of complex numbers from $A$ and $B$. Since $A$ contains 18th roots of unity and $B$ contains 48th roots of unity, the product of any two complex numbers from $A$ and $B$ will be a 144th root of unity. This is because the least common multiple of 18 and 48 is 144. |
| 5. Therefore, the set $C$ contains all 144th roots of unity, which are the complex numbers that satisfy $z^{144} = 1$. Since there are 144th roots of unity, the set $C$ contains 144 distinct elements. |
| So, the final answer is: |
| 144 |

Table 7: Dataset case and LLM output from CommonsenseQA dataset (*Reasoning* Domain)

| Question |
| --- |
| A revolving door is convenient for two direction travel, but it also serves as a security measure at a what? |
| Choices:
(A) bank
(B) library
(C) department store
(D) mall
(E) new york |
| **True Answer** |
| A |
| **LLM Response (After following the instruction in Section C.2.3)** |
| Let's break down the question step by step: |
| 1. The question states that a revolving door is convenient for two-direction travel, which means it allows people to enter and exit the building easily.
2. The question also mentions that the revolving door serves as a security measure, which implies that it provides some kind of barrier or obstacle to prevent unauthorized access.
3. Considering the options, a bank is a likely candidate for a security measure. Banks often have high-security measures in place to protect their customers' money and assets.
4. A library might also have security measures in place, but it's not as likely as a bank. Libraries typically have more relaxed security measures, as they are public spaces.
5. A department store or mall might have some security measures, but they are not as likely to have a revolving door as a security measure. These types of establishments often have more open layouts and may not require as much security.
6. New York is not a type of building or establishment, so it's not a viable option. |
| Based on the reasoning process, the most likely answer is: |
| Answer: A |

Table 8: Dataset case and LLM output from TheoremQA dataset (*Reasoning* Domain)

| **Question** |
| --- |
| Let a undirected graph G with edges E = {<1,2>,<2,4>,<5,4>,<5,6>}, which <A,B> represent Node A is connected to Node B. What is the shortest path from node 1 to node 6? Represent the path as a list. |
| **True Answer** |
| [1, 2, 4, 5, 6] |
| **LLM Response (After following the instruction in Section C.2.3)** |
| To find the shortest path from node 1 to node 6, I will use a breadth-first search (BFS) algorithm. Here's the step-by-step process: |

1. Create a queue and enqueue node 1.
2. Create a set to keep track of visited nodes and add node 1 to it.
3. While the queue is not empty, dequeue a node and explore its neighbors.
4. For each neighbor, check if it's the target node (node 6). If it is, return the path from the starting node to the target node.
5. If the neighbor is not the target node, add it to the queue and mark it as visited.

Here's the BFS traversal:
Queue: [1]
Visited: [1]
Dequeue node 1, explore its neighbors:
Neighbor: 2
Queue: [2]
Visited: [1, 2]

Dequeue node 2, explore its neighbors:
Neighbor: 4
Queue: [4]
Visited: [1, 2, 4]

Dequeue node 4, explore its neighbors:
Neighbor: 5
Queue: [5]
Visited: [1, 2, 4, 5]

Dequeue node 5, explore its neighbors:
Neighbor: 6
Queue: []
Visited: [1, 2, 4, 5, 6]

The target node (node 6) is found. The path from node 1 to node 6 is: [1, 2, 4, 5, 6].

Therefore, the answer is [1, 2, 4, 5, 6].

Table 9: Dataset case and LLM output from MMLU dataset (*Knowledge* Domain)

| Question |
| --- |
| Paper will burn at approximately what temperature in Fahrenheit?

Choices:
(A) 986 degrees
(B) 2125 degrees
(C) 3985 degrees
(D) 451 degrees |
| **True Answer** |
| D |
| **LLM Response (After following the instruction in Section C.2.3)** |
| Let's think step by step!

Paper is a common combustible material, and it's known to burn easily. When we think about the temperatures at which materials burn, we often think of the flash point, which is the lowest temperature at which a substance can ignite and burn.

Paper is a relatively low-temperature material, and it's commonly reported to ignite and burn at around 451°F (233°C). This is often referred to as the "ignition temperature" of paper.

So, considering the options, the correct answer is:

Answer: D |

Table 10: Dataset case and LLM output from Belebele dataset (*Understanding* Domain)

| Question |
| --- |
| "The Irish government is stressing the urgency of parliamentary legislation to rectify the situation. ""It is now important from both a public health and criminal justice perspective that the legislation be enacted as soon as possible"", said a government spokesperson. The Health Minister expressed concern both for the welfare of individuals taking advantage of the temporary legality of the substances involved, and for drug-related convictions handed down since the now-unconstitutional changes came into effect."

Question:
What was temporarily legal in Ireland?

Choices:
(A) Burglary
(B) Fraud
(C) Assault
(D) Drugs |
| **True Answer** |
| D |
| **LLM Response (After following the instruction in Section C.2.3)** |
| Let's break down the passage step by step:

* The passage mentions that the Irish government is stressing the urgency of parliamentary legislation to rectify a situation.
* A government spokesperson is quoted as saying that the legislation needs to be enacted as soon as possible from both a public health and criminal justice perspective.
* The Health Minister is also quoted as expressing concern for the welfare of individuals taking advantage of the temporary legality of the substances involved, and for drug-related convictions handed down since the now-unconstitutional changes came into effect.

From these statements, we can infer that the substances involved are drugs. The passage does not mention any other illegal activities such as burglary, fraud, or assault.

Therefore, the correct answer is:

Answer: D |

## C.2 IMPLEMENTATION

### C.2.1 MODEL SOURCE

All model checkpoints were downloaded from the official HuggingFace[2] repository, with download and license links provided in Table 11. We specifically utilize instruction-based models, as they exhibit superior instruction-following ability and excel in zero-shot generation, making them more aligned with user needs in real-world scenarios.

Table 11: Download and license links of all model artifacts we used in this paper.

| Model Name | Download Link | License Link |
|---|---|---|
| Llama2-7B-Instruct | https://huggingface.co/meta-llama/Llama-2-7b-chat-hf | https://ai.meta.com/llama/license |
| Llama3-8B-Instruct | https://huggingface.co/meta-llama/Meta-Llama-3-8B-Instruct | https://llama.meta.com/llama3/license |
| Qwen1.5-7B-Instruct | https://huggingface.co/Qwen/Qwen1.5-7B-Chat | https://huggingface.co/Qwen/Qwen1.5-7B-Chat/blob/main/LICENSE |
| Qwen2-7B-Instruct | https://huggingface.co/Qwen/Qwen2-7B-Instruct | https://huggingface.co/Qwen/Qwen2-7B-Instruct/blob/main/LICENSE |
| Mistral-7B-Instruct | https://huggingface.co/mistralai/Mistral-7B-Instruct-v0.2 | https://mistral.ai/licenses/MNPL-0.1.md |
| Llama3-70B-Instruct | https://huggingface.co/meta-llama/Meta-Llama-3-70B-Instruct | https://huggingface.co/meta-llama/Meta-Llama-3-70B-Instruct/blob/main/LICENSE |
| Qwen2-72B-Instruct | https://huggingface.co/Qwen/Qwen2-72B-Instruct | https://huggingface.co/Qwen/Qwen2-72B-Instruct/blob/main/LICENSE |

### C.2.2 INFERENCE

Considering the inconsistent difficulty of different tasks, especially since some mathematical tasks may produce longer outputs, we set the **maximum output length to 2048 tokens** and used the `<eos_token>` for truncation. The inference process employs greedy decoding without random sampling. We record the answer completion rate of all models across all datasets (*i.e.*, completing responses within the 2048 output token limit), as shown in Table 12. It can be observed that the models are generally able to complete all responses within this specified length.

Additionally, for a 7B+ model, we deploy it using two 32G V100 GPUs, while for a 70B+ model, we deploy it using four 80G A100 GPUs.

Table 12: Answer completion rate before 2048 limited output token length.

| Dataset 
 Model | MGSM | MATH | CommonsenseQA | TheoremQA | MMLU | Belebele |
|---|---|---|---|---|---|---|
| Llama2-7B-Instruct | 100.00% | 97.98% | 100.00% | 99.25% | 100.00% | 100.00% |
| Llama3-8B-Instruct | 100.00% | 93.48% | 100.00% | 97.75% | 99.82% | 100.00% |
| Qwen1.5-7B-Instruct | 100.00% | 99.70% | 100.00% | 99.88% | 100.00% | 100.00% |
| Qwen2-7B-Instruct | 100.00% | 96.70% | 100.00% | 99.50% | 100.00% | 100.00% |
| Mistral-7B-Instruct | 100.00% | 94.86% | 100.00% | 98.75% | 99.65% | 100.00% |
| Llama3-70B-Instruct | 100.00% | 97.96% | 100.00% | 99.88% | 99.82% | 100.00% |
| Qwen2-72B-Instruct | 100.00% | 99.26% | 99.84% | 98.88% | 99.82% | 100.00% |

### C.2.3 INSTRUCTION

We select instructions followed by LLMs from two open-source projects: OPENCOMPASS[3] and SIMPLE-EVALS[4]. They can ensure the professionalism of instructions. Specifically, all instructions used for each dataset are as follows:

- **MGSM**

> Solve this math problem. Give the reasoning steps before giving the final answer on the last line by itself in the format of "Answer:". Do not add anything other than the integer answer after "Answer:".
> Question:
> `{input_data}`

- **MATH**

> Question: `{input_data}`
> Please reason step by step, and put your final answer within \boxed{}

---

[2] https://huggingface.co/
[3] https://github.com/open-compass/opencompass
[4] https://github.com/openai/simple-evals

- **CommonsenseQA**

> Answer the following multiple choice common-sense reasoning question. The last line of your response should be of the following format: "Answer: $LETTER" (without quotes) where LETTER is one of ABCDE. Think step by step and output the reasoning process before answering.
> `{input_data}`

- **MMLU**

> Answer the following multiple choice question. The last line of your response should be of the following format: "Answer: $LETTER" (without quotes) where LETTER is one of ABCD. Think step by step before answering.
> Question:
> `{input_data}`

- **Belebele**

> Answer the following multiple choice reading-comprehension question. The last line of your response should be of the following format: "Answer: $LETTER" (without quotes) where LETTER is one of ABCD. Please fully understand the passage and give explanations step by step before answering.
> `{input_data}`

- **TheoremQA**

> Below is an instruction that describes a task, paired with an input that provides further context. Write a response that appropriately completes the request.
>
> ### Instruction:
> Please read a math problem, and then think step by step to derive the answer. The answer is decided by Answer Type.
> If the Answer type in [bool], the answer needs to be True or False.
> Else if the Answer type in [integer, float] , The answer needs to be in numerical form.
> Else if the Answer type in [list of integer, list of float] , the answer needs to be a list of number like [2, 3, 4].
> Else if the Answer type in [option], the answer needs to be an option like (a), (b), (c), (d).
> You need to output the answer in your final sentence like 'Therefore, the answer is ...'.
>
> ### Question:
> `{input_data}`
>
> ### Answer_type:
> `{answer_type}`
>
> ### Response:

### C.3 BASELINE

We denote $Y = [y_1, y_2, \cdots, y_T]$ as the output probability distribution including all tokens, and the output logits corresponding to each token are $z_1, z_2, \cdots, z_T$. The model vocabulary is $\mathcal{V}$, so each $y_i$ and $z_i$ are $|\mathcal{V}|$-dimensional vectors, and we have:

$$y_t = \left[ \frac{e^{z_{t1}}}{\sum_{d=1}^{|\mathcal{V}|} e^{z_{td}}}, \frac{e^{z_{t2}}}{\sum_{d=1}^{|\mathcal{V}|} e^{z_{td}}}, \cdots, \frac{e^{z_{t|\mathcal{V}|}}}{\sum_{d=1}^{|\mathcal{V}|} e^{z_{td}}} \right]. \tag{21}$$

The baselines in Section 4.1 are formalized or described as follows:

**1. Verbal Confidence — p(True)** (Kadavath et al., 2022)

p(True) estimates the probability that a model's generation is correct by "asking" the model if its answer is correct. It constructs a new natural language question and takes the likelihood of the next token being True as the uncertainty measure. We follow prompt templates in Kadavath et al. (2022):

```
Question: [...]
Proposed Answer: [...]
Is the proposed answer:
(A) True
(B) False
The proposed answer is:
```

**2. Prompt-Sampling-Aggregation (PSA) Pipeline** (Xiong et al., 2024)

The PSA pipeline involves two progressive steps:

- *Prompt-Sampling*: This step requires generating multiple prompt or question formulations (without changing the original question's semantics) for output sampling. To maximize the guarantee of not using external tools (such as external LLMs rephrasing, manual labeling, etc.), we refer to Gao et al. (2024) to use token-level perturbations by introducing random perturbation characters (like spaces, tabs, etc.) at random positions in the question. This generates $k$ different question inputs for LLMs, resulting in $k$ output texts: $\text{text}_1$, ..., $\text{text}_k$.

- *Aggregation*: After obtaining multiple outputs, it is necessary to measure their consistency. A commonly used measurement method is lexical similarity (Lin et al., 2022c; Kuhn et al., 2023; Chen et al., 2024), which involves using ROUGE-L (Lin, 2004) to assess the similarity of these $k$ outputs in pairs, and then calculating the average as follows:

$$\frac{2}{k(k-1)} \cdot \sum_{i=1}^{k} \sum_{j=i+1}^{k} \text{Rouge} - \text{L}(\text{text}_i, \text{text}_j) \tag{22}$$

We set $k = 5$.

**3. Maximum Softmax Probability** (Si et al., 2022)

Maximum Softmax Probability reflects the maximum probability of the output token probability distribution:

$$\mathbb{E}_{1 \leq t \leq T}\left[\max \boldsymbol{y}_t\right]. \tag{23}$$

• **Perplexity** (Si et al., 2022)

Perplexity reflects the weighted average branching factor of a language:

$$\mathbb{E}_{1 \leq t \leq T}\left[-\log(\max \boldsymbol{y}_t)\right]. \tag{24}$$

• **Entropy** (Huang et al., 2023)

Entropy reflects the distribution uncertainty:

$$\mathbb{E}_{1 \leq t \leq T}\left[\mathbb{E}[-\log \boldsymbol{y}_t]\right]. \tag{25}$$

• **Temperature Scaling** (Shih et al., 2023)

In the softmax operation before obtaining each probability distribution $\boldsymbol{y}_i$, The logit $\boldsymbol{z}_i$ on the exponent is divided by a temperature parameter $T$ to calibrate the final probability. We set $T = 0.7$, the subsequent calculations are consistent with **Maximum Softmax Probability**:

$$\mathbb{E}_{1 \leq t \leq T}\left[\max\left[\frac{e^{z_{t1}/T}}{\sum_{d=1}^{|\mathcal{V}|} e^{z_{td}/T}}, \frac{e^{z_{t2}/T}}{\sum_{d=1}^{|\mathcal{V}|} e^{z_{td}/T}}, \cdots, \frac{e^{z_{t|\mathcal{V}|}/T}}{\sum_{d=1}^{|\mathcal{V}|} e^{z_{td}/T}}\right]\right]. \tag{26}$$

• **Energy** (Liu et al., 2020)

Energy maps logits to an energy equation as a substitute for softmax:

$$\mathbb{E}_{1 \leq t \leq T}\left[-T \cdot \log \sum_{d=1}^{|\mathcal{V}|} e^{z_{td}/T}\right]. \tag{27}$$

We set $T = 0.7$.

• **Monte-Carlo Dropout** (Gal & Ghahramani, 2016)

Monte-Carlo Dropout estimates uncertainty by enabling dropout with different randomness multiple times during the inference phase to obtain multiple output distributions. The variance of these output distributions is used to assess uncertainty. Assuming $k$ random samples yield $k$ outputs $\mathcal{Y} = [\boldsymbol{Y}_1, \boldsymbol{Y}_2, \cdots, \boldsymbol{Y}_k]$, we estimate the uncertainty as follows:

$$\mathbb{E}_{\boldsymbol{Y} \in \mathcal{Y}} \left[ \left( \mathbb{E}_{1 \leq t \leq T_{\boldsymbol{Y}}} \left[ \max \boldsymbol{y}_t \right] - \mathbb{E}_{\boldsymbol{Y} \in \mathcal{Y}} \left[ \mathbb{E}_{1 \leq t \leq T_{\boldsymbol{Y}}} \left[ \max \boldsymbol{y}_t \right] \right] \right)^2 \right]. \tag{28}$$

We set $k = 5$.

• **Length-normalized Entropy** (Malinin & Gales, 2020)

Length-normalized entropy utilizes top-$k$ sampling to generate $k$ outputs $\mathcal{Y} = [\boldsymbol{y}_1, \boldsymbol{y}_2, \cdots, \boldsymbol{y}_k]$. It then computes the average entropy of these outputs as follows:

$$\mathbb{E}_{\boldsymbol{Y} \in \mathcal{Y}} \left[ \mathbb{E}_{1 \leq t \leq T_{\boldsymbol{Y}}} \left[ \mathbb{E}[- \log \boldsymbol{y}_t] \right] \right]. \tag{29}$$

We set $k = 5$.

• **Eigenscore** (Chen et al., 2024)

Eigenscore first performs $k$ decoding sampling, obtaining $k$ embeddings at the $L/2$ layer, along with the covariance matrix $\Sigma$ of these $k$ embeddings. The eigenscore measures uncertainty by calculating the determinant of the covariance $\Sigma$, perturbed by a small addition:

$$\frac{1}{k} \cdot \log \det(\Sigma + \alpha \boldsymbol{I}). \tag{30}$$

We follow (Chen et al., 2024) to set $\alpha = 0.001$ and $k = 5$.

## D ADDITIONAL EXPERIMENTAL RESULTS

### D.1 TASK DIFFICULTY EXPLORATION

Table 9 to 14 presents AUROC results of all seven language models under varying difficulty tasks within the *Mathematics* and *Reasoning* domains.

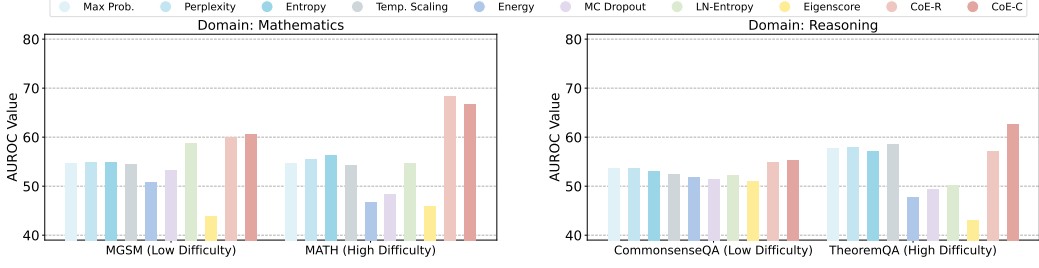

Figure 9: AUROC results (w/ Llama2-7B-Instruct model) of all methods for varying difficulty tasks within the *Mathematics* and *Reasoning* domains.

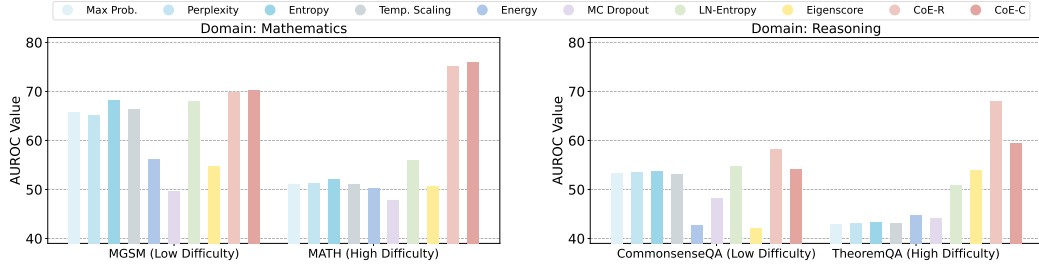

Figure 10: AUROC results (w/ Llama3-8B-Instruct model) of all methods for varying difficulty tasks within the *Mathematics* and *Reasoning* domains.

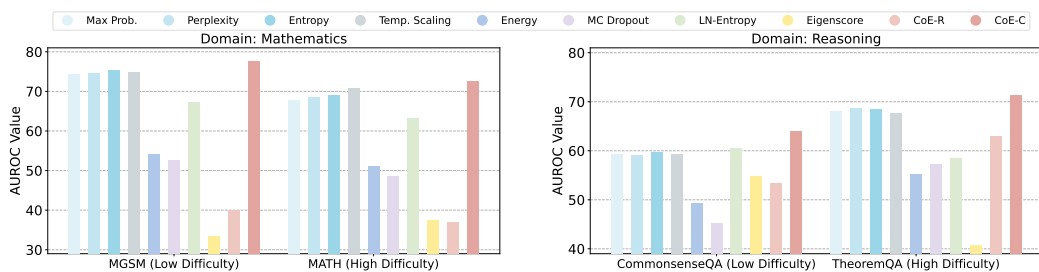

Figure 11: AUROC results (w/ Qwen1.5-7B-Instruct model) of all methods for varying difficulty tasks within the *Mathematics* and *Reasoning* domains.

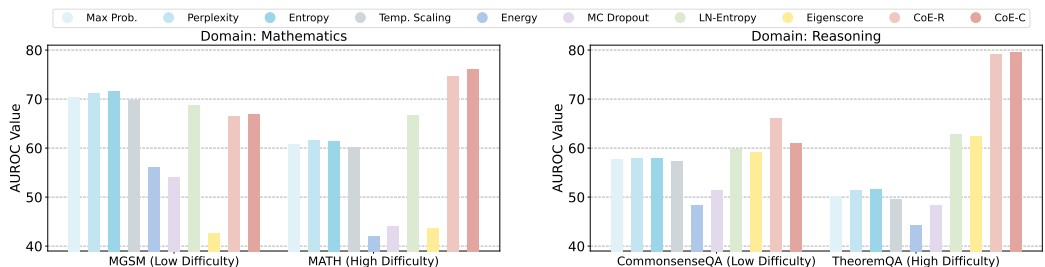

Figure 12: AUROC results (w/ Mistral-7B-Instruct model) of all methods for varying difficulty tasks within the *Mathematics* and *Reasoning* domains.

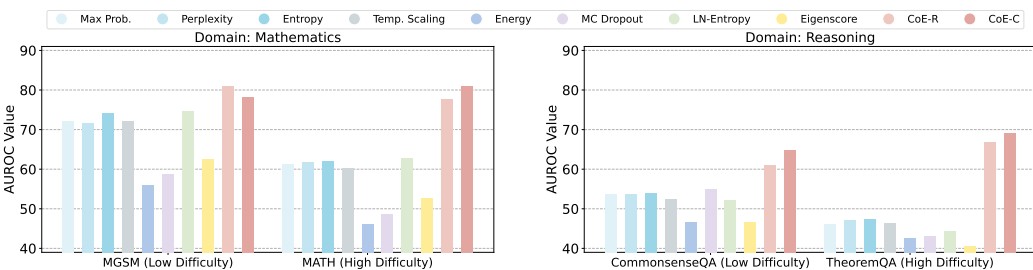

Figure 13: AUROC results (w/ Llama3-70B-Instruct model) of all methods for varying difficulty tasks within the *Mathematics* and *Reasoning* domains.

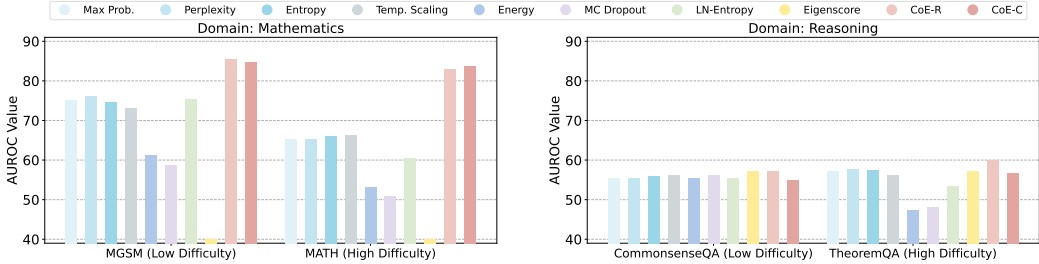

Figure 14: AUROC results (w/ Qwen2-72B-Instruct model) of all methods for varying difficulty tasks within the *Mathematics* and *Reasoning* domains.

## D.2 MULTILINGUAL SCALABILITY

Table 15 - 18 present the AUROC results in four 7B+ language models on the MGSM dataset, which is a mathematical task comprising 11 language versions. The language abbreviations are as follows: Bengali (bn), Chinese (zh), English (en), French (fr), German (de), Japanese (ja), Russian (ru), Spanish (es), Swahili (sw), Telugu (te), and Thai (th).

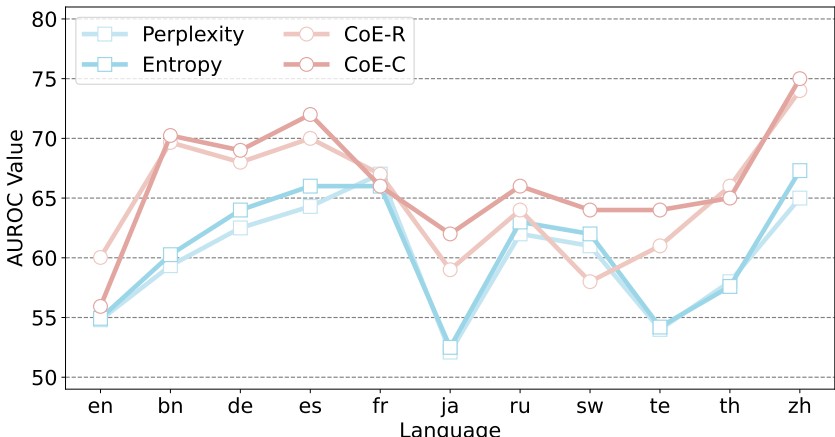

Figure 15: AUROC results (w/ Llama2-7B-Instruct model) on MGSM dataset with 11 languages.

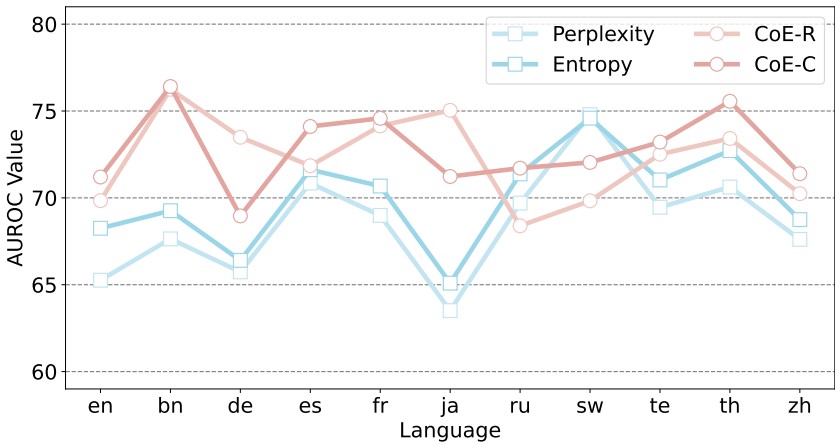

Figure 16: AUROC results (w/ Llama3-8B-Instruct model) on MGSM dataset with 11 languages.

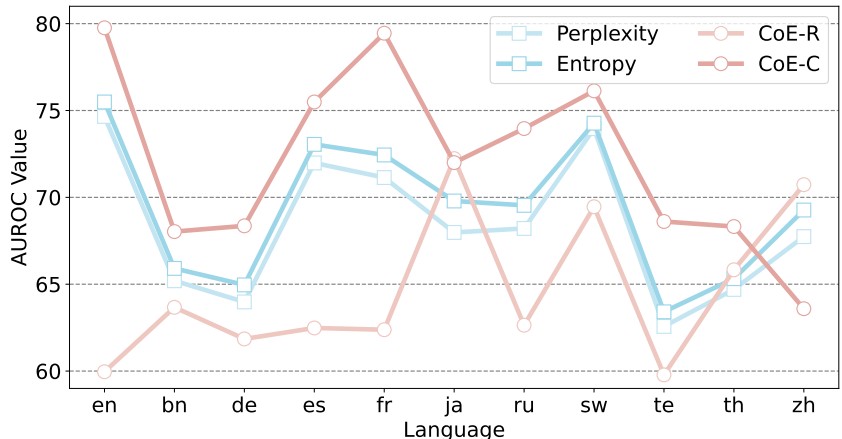

Figure 17: AUROC results (w/ Qwen1.5-7B-Instruct model) on MGSM dataset with 11 languages.

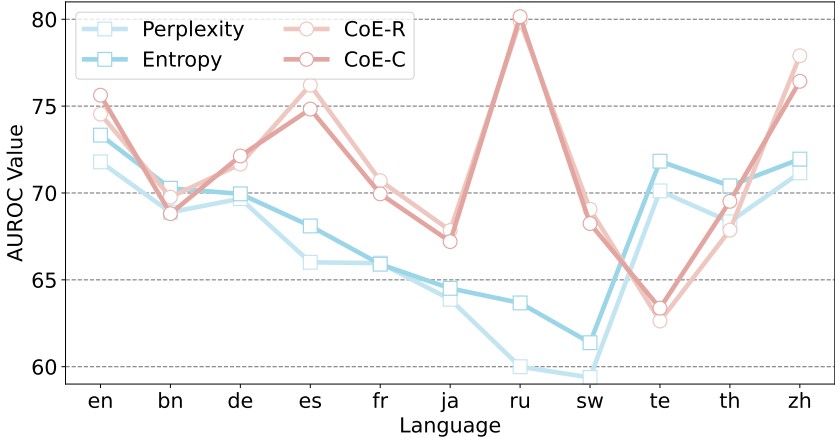

Figure 18: AUROC results (w/ Qwen2-7B-Instruct model) on MGSM dataset with 11 languages.

