# OpenReview forum: "Latent Space Chain-of-Embedding Enables Output-free LLM Self-Evaluation"
_ICLR.cc/2025/Conference — ICLR 2025 Poster_

### Official Review · Reviewer_in4E · 2024-10-27

**Soundness:** 3
**Presentation:** 3
**Contribution:** 2
**Rating:** 6
**Confidence:** 1

**Summary:**

The Chain-of-Embedding (CoE) is a novel approach that allows LLMs to self-evaluate correctness by analyzing their hidden states during inference, forming a latent "thinking path." This label-free method reveals distinct patterns for correct and incorrect responses, enabling real-time, output-free accuracy estimation across various domains. CoE’s minimal computational cost and interpretability make it effective for large-scale applications.

**Strengths:**

1. CoE uses internal hidden states rather than output-based confidence, enhancing adaptability to new tasks without training data.

2. With millisecond-level computation, CoE scales effectively, offering quick feedback for large-scale deployments.

3. CoE provides insights into LLM decision-making by tracing hidden states, offering a transparent, human-like assessment of response correctness.

**Weaknesses:**

1. My major concern is the scenario and the necessity for the label-free scenario. If we limited the LLM query into only math question, commonQA questions, then the label-free scenario may not be necessary. A few-shot human labeling can achieve satisfied performance. The key reason for the label-free is the open-world scenario, which the query can be arbitrary from various domains.  It is impossible for us to label across queries with different purposes. Considering the open-world assumption, there are two following-up question as follows:
1.1. The figure 2 and 3 try to distinguish the discrepancy between the positive and negative samples. When we consider the open-world assumption, there can be more questions with different distributions from a single domain, which may not be able to distinguish.
1.2. The positive samples from different domains show different trajectories, is it possible for the method to distinguish them given queries from different domains

2. The motivation in the introduction part seems confusing. There is no evidence to indicate the correspondence between the human thinking path and LLM intermediate path. Moreover, there may many knowledge-intensive task may not need the thinking path.

3. The evaluation part (especially how to identify the correctness) is not so clear to me. I would like to kindly ask the authors provide a more detailed explanation on it.

4. The path length is related with the number of layer. I think an additional analysis on how the number of layer influence the performance of the proposed method could make the paper more solid.

**Questions:**

1. How does the proposed method handle distributional differences across open-world queries?

2. Can the method reliably distinguish trajectories across diverse domains?

3. What is the evidence for aligning human thinking paths with LLM intermediate paths?

4. How does layer count affect the method's performance?

---

> ### Author Response · Authors · 2024-11-15
> **Response 1: Open-world Application**
>
> Thanks for your constructive feedback. We respond to your questions one by one:
>
> ---
>
> > **W1 & Q1 & Q2**: Scenarios and the necessity for the label-free scenario ... If open-world scenarios: (1) How to handle distributional differences in a single domain? (2) How to distinguish trajectories across diverse domains?
>
> Thank you for your deep reflections on the task setting. This is an open-ended question, as previous work in the field of self-evaluation has been conducted on identically distributed data --- obtaining classifiers on identically distributed data is a common practice. **The open-world assumption you mentioned has not appeared in prior research**, but we are willing to explore this further to validate the generalizability of our work.
>
> > Q1: How does the proposed method handle distributional differences across open-world queries?
>
> We consider the scenario of **different distributions within the same domain** that you mentioned. We **mix GSM8K and MATH datasets (Mathematics Domain) / CommonsenseQA and TheoremQA (Reasoning Domain)**  used in our paper, as they have significantly different data sources and problem difficulties, to simulate this scenario. To ensure data balance, we set the number of samples in the two datasets consistent.
>
> We report AUROC results upon Llama3-8B-Instruct and Qwen2-7B-Instruct models as follows:
>
> *a. Mathematics (GSM8K + MATH)*
>
> || Verbal | PSA | maxprob | ppl | entropy | Temp | Energy | MC Dropout | LN-Entropy | EigenScore | CoE-R(ours) | CoE-C(ours) |
> | --- | --- | --- | --- | --- | --- | --- | --- | --- | --- | --- | --- | --- |
> | Llama3-8B-Instruct | 58.99 | 62.39 | 60.25 | 61.45 | 63.38 | 62.57 | 52.07 | 51.86 | 68.53 | 66.43 | **76.13** | **78.84** |
> | Qwen2-7B-Instruct | 63.94 | 61.03 | 59.19 | 60.32 | 61.35 | 60.56 | 50.39 | 48.56 | 67.72 | 53.34 | **81.40** | **77.94** |
>
> *b. Reasoning (CommonsenseQA + TheoremQA)*
> || Verbal | PSA | maxprob | ppl | entropy | Temp | Energy | MC Dropout | LN-Entropy | EigenScore | CoE-R(ours) | CoE-C(ours) |
> | --- | --- | --- | --- | --- | --- | --- | --- | --- | --- | --- | --- | --- |
> | Llama3-8B-Instruct | 62.92 | 61.25 | 65.46 | 65.97 | 65.68 | 66.24 | 52.30 | 54.16 | 68.82 | 63.39 | **79.93** | **76.94** |
> | Qwen2-7B-Instruct | 57.80 | 65.52 | 66.64 | 67.75 | 67.78 | 65.23 | 57.15 | 50.03 | 66.94 | 66.27 | **75.63** | **77.51** |
>
> It is clear that our method is fully capable of handling scenarios with different distributions within the same domain, and it significantly outperforms other baseline methods. **This validates the generalization of our method under the open-world assumption.**
>
> > Q2: Can the method reliably distinguish trajectories across diverse domains?
>
> Of course, since the trajectory feature differences between different domains are significant, we can cluster any new samples in an open world based on the acquired domain priors. We have verified that the data distributions within the same domain show minimal differences (Q1), so we can obtain the domain prior CoE scores using a small amount of data. After that, we determine which domain's CoE prior score is closest to the CoE score of any new sample to classify its belonging domain. Based on this method, we validate the clustering accuracy:
>
> | Mathematics | Reasoning | Knowledge | Understanding |
> | --- | --- | --- | --- |
> | 95.20 | 96.81 | 99.12 | 97.55|
>
> We find that all accuracies are higher than 95%, indicating that **our method can effectively distinguish trajectories from different domains**.

---

> ### Author Response · Authors · 2024-11-15
> **Response 2: Motivation**
>
> > **W2 & Q3**: There is no evidence to indicate the correspondence between the human thinking path and LLM intermediate path. Moreover, there may many knowledge-intensive task may not need the thinking path.
>
> Thank you for pointing this out, we will clarify our motivations as follows:
>
> > What is the evidence for aligning human thinking paths with LLM intermediate paths?
>
> We do not need to provide evidence for the alignment of human and LLM thinking paths, as **all descriptions of human thinking paths are *heuristic* for our LLM research**. The logic of our motivation is as follows:
>
> ● Firstly, we claim that there are differences in the thinking path of the human brain when considering right and wrong. We want to draw an analogy to assume that *"LLM thinking paths also exhibit differences"*. **Our focus in this analogy is on the behavior of "exhibiting differences" rather than on the concept of "thinking path"**. (The fundamental logic here is: This analogy assumption does not need to be based on the alignment of the thinking paths of humans and LLMs.)
>
> ● Based on this analogy assumption, **we only need to define the thinking path of the LLM and verify this assumption.** As stated in the introduction, the LLM models syntax at a low level and semantics at a high level, which constitutes a latent thinking path.
>
> Overall, our motivation is heuristic, focusing on the "differences in thinking paths." The core of our paper is to validate the differences in thinking paths in LLMs. This does not need to be grounded in the alignment between LLMs and humans, nor does it require validating the compatibility between the two through the conclusions of the paper. In fact, **Both Reviewer 2nBf and GkSP recognize the rationality of our motivation and affirm this strength.**
>
> > Moreover, there may many knowledge-intensive task may not need the thinking path.
>
> **I believe you have confused our concept of "latent thinking paths" with "Chain-of-Thought (CoT)"**, which is a post-hoc thinking process that is reflected in the model's output rather than within the model itself. From the view of LLM-generated answers, responding to such questions indeed may not require specific steps, but **this feature claimed by you points to the concept of CoT.**
>
> As for latent thinking paths, even for knowledge-based tasks, there will certainly be steps involving modeling syntax, grammar, semantic information, and memory retrieval. Although it may not be clearly explainable, these latent processing definitely exists.

---

> ### Author Response · Authors · 2024-11-15
> **Response 3: Other Clarification**
>
> > **W3**: A more detailed explanation on how to identify the correctness
>
> Happy to address your question, and I pinpoint it to line 320. In fact, we have aligned with the evaluation framework that OpenAI used when evaluating the GPT series models (see **Appendix C.2.3, line 1442**, https://github.com/openai/simple-evals). I will use the MGSM dataset as an example to explain how to identify the correctness of LLM responses (i.e., how to extract the ground-truth correctness labels).
>
> Due to what we used being instruct-based models, we prompt LLMs to follow specific instruction formats when generating answers, which facilitates our answer extraction. For example:
>
> ```python
> Solve this math problem. Give the reasoning steps before giving the final answer on the last line by itself in the format of "Answer:". Do not add anything other than the integer answer after "Answer:".
> Question: {input_data}
> ```
> When LLMs receive the above instructions and questions, it will follow the format to generate the following contents:
> ```python
> [......]
> Answer: [...]
> ```
> Now, we only need to use regular expression matching to extract the content that follows "Answer: " to obtain the exact answer generated by the LLM. If it matches the true label, the correctness label is set to 1; otherwise, it is set to 0. All instructions have been presented in Appendix C.2.3.
>
>
> ---
>
> >  **W4 & Q4**:  Experiments --- Layer number analysis
>
> I agree with your viewpoint that since there is a correlation between the CoE and the layer number, it is meaningful to analyze the impact of the layer number on performance.
>
> In fact, among the seven LLMs we selected, **the influence of the layer numbers on performances can already be concluded in Table 1** due to the differentiation in parameter scales. We label the layer numbers of these LLMs and copy the AUROC results of the CoE-R metric as follows:
>
> | | Llama2-7B-Instruct (32 layers) | Llama3-8B-Instruct (32 layers) | Qwen1.5-7B-Instruct (28 layers)  | Qwen2-7B-Instruct (28 layers) | Mistral-7B-Instruct (32 layers) | Llama3-70B-Instruct (80 layers) | Qwen2-72B-Instruct (80 layers) |
> | --- | --- | --- | --- | --- | --- | --- | --- |
> | Mathematics | 63.63 | 73.08 | 77.22 | 76.68 | 72.24 | **79.35** | **84.34** |
> | Reasoning | 59.00 | 55.85 | 67.67 | 62.70 | 70.79 | 66.93 | 61.86 |
> | Knowledge | 59.07 | 62.45 | 62.11 | 61.85 | 62.18 | **66.41** | **73.15** |
> | Understanding | 55.49 | 58.47 | 55.11 | 70.87 | 66.70 | **73.32** | **74.88** |
>
> * The first five LLMs are all 7B+ models with 28/32 layers; the last two LLMs are both 70B+ models with 80 layers:
>
>    * From the results, **on 3/4 of the domains, LLMs with more layers (80 layers) significantly outperform those with fewer layers (30 layers)**. Although we cannot definitively conclude that "more layers always mean better performance," there is indeed a noticeable trend.
>    * This trend is quite reasonable: as the number of layers increases, the amount of information contained within the model grows, allowing for more features in the trajectory to effectively distinguish the correct samples.
>    * Based on this analysis, **our method shows promise: As the demand for larger-scale LLMs (more parameters and more layers) surges in the industry, the enhanced model scaling robustness allows our method for widespread deployment in real-world scenarios, ensuring its broad generalizability.**
>
> ---
>
> Additionally, this work serves as a foundation that can be extended to broader applications, such as decoding and preference optimization **(Refer to our discussion with Reviewer 2nBf)**.
>
> Finally, thank you once again for taking the time to review our work and provide valuable insights. We hope our response can address your concerns and that you can recognize the value of our work. We look forward to your more positive feedback.

---

> > ### Comment · Reviewer_in4E · 2024-11-15
> >
> > Thanks for your response. I have raised my score accordingly

---

> > > ### Author Response · Authors · 2024-11-17
> > > **Thanks for increasing the score**
> > >
> > > Thank you very much for your timely response and for increasing the score. If you have any further questions, we would be happy to discuss them.

---

### Official Review · Reviewer_GkSP · 2024-11-03

**Soundness:** 2
**Presentation:** 3
**Contribution:** 2
**Rating:** 6
**Confidence:** 3

**Summary:**

The authors propose a metric for label-free LLM self-evaluation, that utilizes the observed discrepancies in progressive hidden states when LLMs generate correct and incorrect responses. Intuited by the cognitive phenomenon in human thinking, the authors measure the CoE discrepancy in the two sets, responding correctly and incorrectly. Based on the obvious discrepancies in Magnitude and Angle, the authors give the two metrics, CoE-R and CoE-C. Finally, the authors achieved SOTA in various datasets and backbone LLMs. Furthermore, there are other analyses in many aspects, e.g., efficiency and multilingual scalability.

**Strengths:**

1. The literature of this paper is comprehensive.

2. The exploration is structured and natural. From intuitive cognitive phenomenon to observation, the authors verify the assumpted discrepancy between correct and incorrect generation. Then, the authors proposed a metric and then demonstrated the effectiveness of the metric. The idea is well-founded.

3. The authors verify their proposed metrics on various backbones and datasets.

4. The authors give a detailed analysis of many aspects.

**Weaknesses:**

1. Missing critical statistic analysis of discrepancies. The foundation of the proposed metrics is based on the existence of the discrepancy between correct and incorrect generation. However, only visualizing this discrepancy is not enough.

2. Why is the targeted task self-evaluation? self-evaluation is a task that seems appealing but is quite ambiguous and questionable; even if the reviewer reads the reference paper cited by authors for self-evaluation, there is no task definition of self-evaluation.

In the authors' presentation, based on the experiment setting and references cited by the authors, self-evaluation is highly associated with uncertainty estimation, but the authors don't mention their targeted method for uncertainty estimation; furthermore, the authors present self-evaluation as a potential method enabling evaluating LLM's responses without labels in the introduction section, which is associated with llm-as-a-judge and self-rewarding. There is a big gap between the two tasks. For label-free self-evaluation, I advise authors to compare their metrics with llm-as-a-judge or self-rewarding methods (label-based self-evaluation) in some instruction-following tasks, even not sota.

3. Missing verifying on some key datasets, for example, triviaQA and TruthfulQA, which have been tested in many previous works cited by authors;

4. Missing critical robustness analysis on adversarial samples, especially uncertainty estimation.

**Questions:**

a. How does this metric work? Is there a threshold to decide the classification using metrics? If so, what is this threshold?

b. In line 104, why does the average embedding at layer l represent the l-th sentence hidden state?

**Details Of Ethics Concerns:**

No ethics review is needed.

---

> ### Author Response · Authors · 2024-11-15
> **Response 1: Targeted Task Choice**
>
> Thanks for your constructive feedback. We respond to your questions one by one:
>
> ---
>
> > **W2:** Why is the targeted task self-evaluation? self-evaluation is a task that seems appealing but is quite ambiguous and questionable; even if the reviewer reads the reference paper cited by authors for self-evaluation, there is no task definition of self-evaluation... (1) Self-evaluation is highly associated with uncertainty estimation... (2) Self-evaluation is associated with llm-as-a-judge and self-rewarding...
>
> Thank you for your thorough reading and deep reflection on our paper. Regarding the self-evaluation setting, this is a point worth discussing, and we organize our views as follows:
>
> > The definition of self-evaluation is ambiguous and questionable
>
> The definition of self-evaluation is a relatively ambiguous concept, and previous papers indeed do not completely give its definition.
> First, from [1], we can identify the nascent concept of self-evaluation [2], whose ***research objective* is to "explore whether LLMs are aware of the correctness of their responses."** Considering the ambiguity of the definition, we can only integrate the method categories that align with self-evaluation research objectives, thereby forming a methodological framework for self-evaluation:
>
> * The related work in [3] comprehensively delineates the scope of methods involved in self-evaluation, including Verbal Confidence and the sampling-based Prompt-Sampling-Aggregation (PSA) Pipeline.
> * The introduction in [4] also clearly categorizes methods based on logits/probabilities as a category under self-evaluation, which actually aligns closely with the traditional scope of uncertainty estimation that you mentioned.
> Based on these backgrounds, we integrate a framework of methods that aligns with the research objectives of self-evaluation, and categorize according to characteristics:
>
> Based on these backgrounds, we integrate a framework of methods that aligns with the research objectives of self-evaluation, and categorize according to characteristics:
>
> | | black/white-box | sampling-based | access output logits | access hidden states |
> | --- | --- | --- | --- | --- |
> | 1. Verbal Confidence | black | × | × | × |
> | 2. PSA pipeline | black | √ | × | × |
> | 3. Uncertainty Estimation | white | × | √ | × |
>
> Note that these three categories are aligned with our introduction (paraphrase 2) and the experimental baselines.
>
> > Why is the targeted task self-evaluation?
>
> **Our work aligns with the research objectives of self-evaluation, but it does not belong to any of the three method categories mentioned above.** Therefore, we tend to define our study under the concept of self-evaluation and discuss and compare it with other method categories that have the same research objective.
>
> > Self-evaluation is highly associated with uncertainty estimation. The authors don't mention their targeted method for uncertainty estimation.
>
> As we integrated above, **our method and uncertainty estimation methods are parallel, both serving the same research objective** of measuring the likelihood of "LLM answering correctly" through a scalar score. However, our method falls into the fourth category:
>
> | | black/white-box | sampling-based | access output logits | access hidden states |
> | --- | --- | --- | --- | --- |
> | 4. our CoE method | white | × | × | √ |
>
> Indeed, our CoE method is closest to the uncertainty estimation for we are both white-box methods, but we cannot categorize our method within the realm of uncertainty estimation for we access different components inside the LLMs.
>
> > Self-evaluation is associated with llm-as-a-judge and self-rewarding. ... For label-free self-evaluation, I advise authors to compare their metrics with llm-as-a-judge or self-rewarding methods (label-based self-evaluation) in some instruction-following tasks, even not sota.
>
> Thanks a lot for your kind suggestion, but I’m sorry that **self-evaluation is completely contradictory with LLM-as-a-judge and label-based evaluation.**
>
> To be simplest, we just need to go back to the self-evaluation research objective[2] of "explore whether LLMs are aware of the correctness of their responses." — this means that **our research focus is on the LLMs themselves**. For LLM-as-a-judge, they investigate whether an external LLM can be used to judge the LLM of our study, so **their research focus is on external LLMs**; and for label-based evaluation, it is completely unnecessary to consider the LLM's own awareness of what it knows. Therefore, self-evaluation and the two types of research you mentioned are not on the same track.

---

> > ### Author Response · Authors · 2024-11-15
> >
> > **Overall, we start from the research goal of self-evaluation to integrating all relevant method categories. The characteristics of our method result in our research not fitting into any existing method category, so we have integrated all of these categories for joint discussion and comparison.**
> >
> > We hope our explanation provides you with a clearer understanding of our research goals and scope. If you have any new questions about this weakness, feel free to discuss them with us.
> >
> > ---
> >
> > [1] Self-Evaluation Guided Beam Search for Reasoning. NeurIPS 2023.
> >
> > [2] Language Models (Mostly) Know What They Know.
> >
> > [3] Think Twice Before Trusting: Self-Detection for Large Language Models through Comprehensive Answer Reflection. EMNLP 2024.
> >
> > [4] Self-Evaluation Improves Selective Generation in Large Language Models.

---

> ### Author Response · Authors · 2024-11-15
> **Response 2: Additional Results (1)**
>
> > **W1:** Missing critical statistic analysis of discrepancies.
>
> Thank you for pointing this out. In Section 2, **Figure 2 has already represented the quantitative data analysis you mentioned**, while Figure 3 is the qualitative visualization analysis. In Figure 2, we present the statistical data (i.e., the Magnitude and Angle values of all samples) **in the form of a two-dimensional distribution, only allowing readers to intuitively grasp the patterns behind the data.**
>
> Of course, we appreciate your rigorous considerations, so we report all data points from Figure 2 (including the mean and standard deviation of each feature) as follows:
>
> *Feature 1: Magnitude*
>
> || Mathematics | Reasoning | Knowledge | Understanding |
> |---|---|---|---|---|
> | Correct Samples|0.180 ± 0.015|0.164 ± 0.015|0.441 ± 0.103|0.174 ± 0.016|
> |Incorrect Samples|0.159 ± 0.010|0.151 ± 0.011|0.259 ± 0.075|0.148 ± 0.010|
>
> *Feature 2: Angle*
>
> || Mathematics | Reasoning | Knowledge | Understanding |
> |---|---|---|---|---|
> |Correct Samples|0.179 ± 0.010|0.175 ± 0.011|0.154 ± 0.018|0.160 ± 0.008|
> |Incorrect Samples|0.194 ± 0.014|0.189 ± 0.014|0.169 ± 0.019|0.181 ± 0.009|
>
> **The statistics are consistent with the patterns presented in Figure 2.** I hope our clarification can help you eliminate any misunderstanding you have regarding this weakness.
>
> > **W3**: Experiments --- triviaQA and TruthfulQA datasets
>
> Thanks for your mention. We report AUROC results on the two datasets upon Llama3-8B-Instruct and Qwen2-7B-Instruct models.
>
> *(a) triviaQA*
>
> || Verbal | PSA | maxprob | ppl | entropy | Temp | Energy | MC Dropout | LN-Entropy | EigenScore | CoE-R(ours) | CoE-C(ours) |
> | --- | --- | --- | --- | --- | --- | --- | --- | --- | --- | --- | --- | --- |
> |Llama3-8B-Instruct|61.02|62.79|58.56|58.94|59.21|57.72|43.15|48.62|59.63|68.96|68.92|**69.94**|
> |Qwen2-7B-Instruct|59.41|56.38|61.17|62.25|62.39|61.95|64.42|45.25|64.79|60.08|**73.25**|**72.17**|
>
> *(b) TruthfulQA*
>
> || Verbal | PSA | maxprob | ppl | entropy | Temp | Energy | MC Dropout | LN-Entropy | EigenScore | CoE-R(ours) | CoE-C(ours) |
> | --- | --- | --- | --- | --- | --- | --- | --- | --- | --- | --- | --- | --- |
> |Llama3-8B-Instruct|59.94|62.24|65.71|66.09|66.58|62.57|53.38|46.87|68.27|67.62|**72.21**|**74.74**|
> |Qwen2-7B-Instruct|58.21|56.38|62.20|62.20|62.56|60.09|52.23|50.16|67.52|63.39|**75.18**|**76.09**|
>
> **From the results, our method still demonstrates excellent performance on these two datasets.**

---

> ### Author Response · Authors · 2024-11-15
> **Response 2: Additional Results (2)**
>
> > **W4:** Experiments --- Robustness analysis on adversarial samples
>
> Thank you for your consideration. To the best of our knowledge, **the previous self-evaluation work did not take into account adversarial robustness, which is not a well-defined problem in this research area**. Upon reflection, we realize that **perturbing the samples might affect the accuracy of the LLM's responses, which would change the ratio of positive to negative samples**. Based on this, we can even treat the evaluation of the perturbed data as a completely new evaluation setting on a separate set of independent data.
>
> Of course, we fully support your concerns for it is important for real-world employment. Therefore, we refer to [1] to construct two batches of perturbation data, with the following construction methods:
>
> * Paraphrasing: We generate one paraphrased input by querying ChatGPT using the prompt in [1];
> * Dummy Tokens: We randomly select tokens that marginally influence the original meaning and append them to the input. Such tokens could be newline characters, tab spaces, ellipses, or supplementary punctuation marks.
>
> We compare two settings:
>
> * Original: Results in our paper
> * Perturbation: We replace raw samples in the dataset with the perturbed data. We report results under two perturbations.
>
> We report AUROC results upon Llama3-8B-Instruct and Qwen2-7B-Instruct models as follows:
>
> ---
>
> **(a) Llama3-8B-Instruct**
>
> *Domain I: Mathematics*
>
> ||Perplexity|Entropy|LN-Entropy|CoE-R(ours)|CoE-C(ours)|
> |---|---|---|---|---|---|
> |Original|58.32|60.17|61.95|72.54|73.08|
> |Perturbation w/ Paraphrasing|56.24|60.02|55.93|72.68|74.51|
> |Perturbation w/ Dummy Tokens|55.48|57.89|60.39|71.36|72.26|
>
> *Domain II: Reasoning*
>
> |    | Perplexity | Entropy | LN-Entropy | CoE-R(ours) | CoE-C(ours)  |
> |------|-----|--|---|-----|----|
> | Original      | 48.40       | 48.56   | 52.88      | 63.12       | 55.85        |
> | Perturbation w/ Paraphrasing | 49.32      | 50.60    | 52.01      | 63.29       | 56.39        |
> | Perturbation w/ Dummy Tokens | 44.82 | 48.26   | 49.43      | 62.71       | 57.25        |
>
> *Domain III: Knowledge*
>
> |  | Perplexity | Entropy | LN-Entropy | CoE-R(ours) | CoE-C(ours)  |
> |---|---|--|--|-----|---|
> | Original   | 50.50  | 50.12   | 58.30 | 64.20  | 62.45   |
> | Perturbation w/ Paraphrasing | 52.34      | 47.62   | 55.43      | 64.39       | 61.84        |
> | Perturbation w/ Dummy Tokens | 48.52      | 46.34   | 58.02      | 62.95       | 62.17        |
>
> *Domain IV: Understanding*
>
> |  | Perplexity | Entropy | LN-Entropy | CoE-R(ours) | CoE-C(ours)  |
> |----|---|---|-----|--|--|
> | Original  | 56.64  | 56.78   | 55.42  | 64.81  | 58.47  |
> | Perturbation w/ Paraphrasing | 56.62  | 56.08   | 55.69  | 64.73  | 57.86   |
> | Perturbation w/ Dummy Tokens | 56.12  | 55.76   | 53.21 | 64.52| 58.68   |
>
> ---
>
> **(b) Qwen2-7B-Instruct**
>
> *Domain I: Mathematics*
>
> | | Perplexity | Entropy | LN-Entropy | CoE-R(ours) | CoE-C(ours)  |
> |--|---|--|--|--|-|
> | Original    | 58.83 | 70.25   | 66.63 | 75.75| 76.68  |
> | Perturbation w/ Paraphrasing | 54.39  | 70.10    | 62.53      | 74.56       | 76.20 |
> | Perturbation w/ Dummy Tokens | 55.75  | 64.38   | 63.35      | 75.27       | 76.13  |
>
> *Domain II: Reasoning*
>
> | | Perplexity | Entropy | LN-Entropy | CoE-R(ours) | CoE-C(ours)  |
> |--|--|-|---|--|---|
> | Original                     | 55.39      | 55.97   | 56.07  | 66.68       | 62.70   |
> | Perturbation w/ Paraphrasing | 53.34      | 54.68   | 57.12      | 66.30   | 61.59        |
> | Perturbation w/ Dummy Tokens | 52.06      | 50.13   | 54.48      | 66.22   | 62.43        |
>
> *Domain III: Knowledge*
>
> | | Perplexity | Entropy | LN-Entropy | CoE-R(ours) | CoE-C(ours)  |
> |-----|--|---|--|-|---|
> | Original  | 57.26   | 57.80   | 58.01      | 66.68       | 62.70         |
> | Perturbation w/ Paraphrasing | 52.69      | 56.23   | 52.36      | 67.25       | 62.67        |
> | Perturbation w/ Dummy Tokens | 54.38      | 53.67   | 52.69      | 65.39       | 62.03        |
>
>
> *Domain IV: Understanding*
>
> | | Perplexity | Entropy | LN-Entropy | CoE-R(ours) | CoE-C(ours)  |
> |--|-|---|-|--|-|
> | Original  | 60.93   | 62.65   | 63.59  | 71.92  | 70.87 |
> | Perturbation w/ Paraphrasing | 60.08  | 60.26  | 59.36 | 71.56  | 71.50   |
> | Perturbation w/ Dummy Tokens | 58.67  | 60.15  | 62.37| 71.79  | 70.49    |
>
> ---
>
> **After perturbing the raw data, the performance of our method remains stable, and is better than other uncertainty estimation baselines.** We find that compared with our method, some uncertainty estimation methods, such as PPL, exhibit an overconfidence phenomenon in LLMs. When faced with perturbations, LLMs may produce incorrect answers for samples that were originally answered correctly, yet their output probabilities remain relatively high.
>
> **These results and conclusions indicate that our method exhibits sufficient robustness against adversarial perturbations.**
>
> [1] SPUQ: Perturbation-Based Uncertainty Quantification for Large Language Models. EACL, 2024.

---

> ### Author Response · Authors · 2024-11-16
> **Response 3: Other Questions**
>
> > **Q1:** Is there a threshold to decide the classification using metrics?
>
> Yes, we can use the AUROC curve to obtain a threshold, this is a standard way as prior work. We refer to [1]: "We computer the optimal cut-off $\tau_i$ of **Youden Index**, which is **at the point in the AUROC curve where TPR-FPR is maximum**." (The Youden Index is a standard statistical measure used to evaluate the performance of classifiers.)
>
> After obtaining the threshold $\tau_i$, for each sample, **if the metric score (CoE or other baselines) is greater than $\tau_i$, it is classified as correct; otherwise, it is classified as incorrect.** Based on this criterion, we can derive a threshold after obtaining an AUROC curve for each dataset/model/metric, and then calculate the **Accuracy**.
>
> We report Accuracy results upon Llama3-8B-Instruct and Qwen2-7B-Instruct models as follows:
>
> **Domain I: Mathematics**
>
> |  | Verbal | PSA  | maxprob | ppl   | entropy | Temp | Energy | MC Dropout | LN-Entropy | EigenScore | CoE-R(ours) | CoE-C(ours)  |
> |---|----|--|--|--|---|------|---|------|--|----|-----|----|
> | Llama3-8B-Instruct | 62.40   | 54.40 | 69.60    | 72.62 | 75.20    | 71.60 | 56.80   | 61.60       | 63.60       | 46.40       | **80.00**          | **80.40**  |
> | Qwen2-7B-Instruct  | 47.60   | 55.20 | 63.60    | 60.40  | 64.80    | 64.80 | 55.20   | 53.60       | 67.60       | 64.80       | **82.80**        | **68.00**  |
>
> **Domain II: Reasoning**
>
> |   | Verbal | PSA    | maxprob | ppl    | entropy | Temp   | Energy | MC Dropout | LN-Entropy | EigenScore | CoE-R(ours) | CoE-C(ours)  |
> |-----|--------|--------|---------|--------|---------|--------|--------|------------|------------|------------|-------------|--------------|
> | Llama3-8B-Instruct | 54.02  | 58.95  | 51.36   | 54.93  | 52.03   | 57.14  | 48.92  | 46.24      | 59.66      | 61.14      | 72.38       | 67.39        |
> | Qwen2-7B-Instruct  | 49.32  | 57.10  | 61.75   | 64.25  | 67.63   | 68.51  | 60.27  | 51.01      | 62.41      | 58.39      | 70.63       | 72.12        |
>
> **Domain III: Knowledge**
>
> |     | Verbal | PSA    | maxprob | ppl    | entropy | Temp   | Energy | MC Dropout | LN-Entropy | EigenScore | CoE-R(ours) | CoE-C(ours)  |
> |---|----|----|----|---|-----|---|---|----|------|----|-----|------|
> | Llama3-8B-Instruct | 62.36  | 53.68  | 61.05   | 60.53  | 59.82   | 64.09  | 58.66  | 56.27      | 63.15      | 61.05      | 69.12       | 68.33        |
> | Qwen2-7B-Instruct  | 60.36  | 58.32  | 51.75   | 51.75  | 53.16   | 54.10  | 48.96  | 47.23      | 64.24      | 56.28      | 63.62       | 68.60        |
>
> **Domain IV: Understanding**
>
> |    | Verbal | PSA    | maxprob | ppl    | entropy | Temp   | Energy | MC Dropout | LN-Entropy | EigenScore | CoE-R(ours) | CoE-C(ours)  |
> |----|----|---|--|-----|---|----|----|-----|----|---|------|----|
> | Llama3-8B-Instruct | 51.36  | 54.20  | 61.24   | 57.19  | 58.62   | 56.62  | 58.91  | 50.09      | 63.98      | 62.14      | 70.16       | 72.32        |
> | Qwen2-7B-Instruct  | 54.23  | 51.09  | 66.98   | 69.20  | 62.18   | 63.67  | 55.53  | 49.93      | 70.68      | 71.05      | 78.65       | 77.09        |
>
>
> **After obtaining the threshold and calculating the classification accuracy, our method still maintains optimal performance.**
>
> [1] Embedding Trajectory for Out-of-Distribution Detection in Mathematical Reasoning. NeurIPS 2024.
>
> ---
>
> > Q2: Why does the average embedding at layer l represent the l-th sentence hidden state?
>
> Regarding the representation of the sentence hidden state, we **strictly align with the definitions in the previous papers**, as shown below:
>
> * [1] ... To obtain the output embedding, we average the decoder’s final-layer hidden state vectors.
> * [2] ... the sentence embedding can be obtained by averaging the token embedding ...
> * [3] ... we define the average embedding as the sentence embedding at layer l.
>
> A reasonable explanation is that each token in a sentence contributes to the overall semantics of the sentence. In fact, in traditional sentence embedding research, this approach has already existed as a naive modeling way[4,5].
>
> [1] Out-of-Distribution Detection and Selective Generation for Conditional Language Models. ICLR 2023.
>
> [2] INSIDE: LLMs' Internal States Retain the Power of Hallucination Detection. ICLR 2024.
>
> [3] Embedding Trajectory for Out-of-Distribution Detection in Mathematical Reasoning. NeurIPS 2024.
>
> [4] SimCSE: Simple Contrastive Learning of Sentence Embeddings. EMNLP 2021.
>
> [5] Sentence-BERT: Sentence Embeddings using Siamese BERT-Networks. EMNLP 2019.
>
> ---
>
> Additionally, this work serves as a foundation that can be extended to broader applications, such as decoding and preference optimization **(Refer to our discussion with Reviewer 2nBf)**.
>
> Finally, thank you once again for taking the time to review our work and provide valuable insights. We hope our response can address your concerns and that you can recognize the value of our work. We look forward to your more positive feedback.

---

> ### Author Response · Authors · 2024-11-21
> **Look forward to feedbacks**
>
> Dear Reviewer GkSP,
>
> We first thank you again for your constructive comments. We have addressed your concerns one by one and supplemented more detailed experimental results and explanations. We look forward to further discussion with you and your positive feedback about our rebuttal.
>
> Best regards,
>
> Authors

---

> > ### Comment · Reviewer_GkSP · 2024-11-25
> >
> > Since the authors have addressed most of my concerns, I would like to raise the score.

---

> > > ### Author Response · Authors · 2024-11-26
> > >
> > > We are happy to have addressed your concerns and thank you for raising the score.

---

### Official Review · Reviewer_2nBf · 2024-11-06

**Soundness:** 3
**Presentation:** 3
**Contribution:** 3
**Rating:** 6
**Confidence:** 3

**Summary:**

The authors propose the Chain-of-Embedding (CoE) in the latent space to enable LLMs to perform output-free self-evaluation, and find that when LLMs respond correctly and incorrectly, their CoE features differ, these discrepancies assist us in estimating LLM response correctness.

**Strengths:**

It is quite interesting to discover that the hidden states of LLMs can be utilized to estimate the LLM response correctness without any label, which inspires me a lot.

Moreover, it is also reasonable to treat the progressive hidden states as the latent thinking path of LLMs, leading to the following
assumption: CoE discrepancies may happen when LLMs generate correct and incorrect responses.

Overall, it is quite inspireable paper and give the readers a lot of insights on the thinking system of LLMs

**Weaknesses:**

I donot think this paper have any significant weaknesses

**Questions:**

My only question is,  there are a a lot of interesting findings in this paper, such as
1. Hidden states to estimate the LLM response correctness without any label
2. CoE discrepancies may happen when LLMs generate correct and incorrect responses,
how can these findings be used to improve the reasoning accuracy of LLMs during pre-training or post-training?

---

> ### Author Response · Authors · 2024-11-15
> **Discussion about future application**
>
> > how can these findings be used to improve the reasoning accuracy of LLMs during pre-training or post-training?
>
> First, thanks very much for your recognition of our work. Regarding your only question, we would also be happy to discuss the future application value of the CoE with you. From our view, we think that these findings can be used to improve the reasoning accuracy of LLMs from the following topics:
>
> **(1) Inference-time Decoding**
>
> In this work, we find that CoE can serve as a measure of the latent thinking paths of LLMs and can distinguish between correct and incorrect samples. This allows us to apply it to decoding sampling during inference time: similar to self-consistency[1], but it only considers voting on the exact answers to the output text. CoE, on the other hand, brings in more potential information from within the model, enabling us to sample multiple trajectories and select the most suitable one as the final voting result.
>
> **(2) Preference Optimization**
>
> Since we can use CoE to obtain a score (CoE-R and CoE-C) that reflects the correctness of the LLM responses, we can use it as a scorer for preference data pairs, thereby enabling self-iterative preference optimization in the post-training phase. This idea of self-iterative PO is similar to a recent work[2], but it demonstrates a significantly higher efficiency using single trajectories rather than multiple sampling.
>
> In general, this paper can serve as an inspirational study for interpretable mechanisms that have many application values in the future. There must be more valuable scenarios waiting to be explored. We welcome your suggestions and more positive feedback.
>
> [1] Self-Consistency Improves Chain of Thought Reasoning in Language Models. ICLR 2023.
>
> [2] Self-Consistency Preference Optimization.

---

> > ### Comment · Reviewer_2nBf · 2024-11-16
> >
> > Thanks, from my view, these two aforementioned methods are not so exciting to me. What I am really interested in is how to improve the optimization method rather than the data selection.
> >
> > Overall, I would like to keep my score, thanks for your efforts

---

### Author Response · Authors · 2024-11-24
**Global Response and Rebuttal Summary**

We would like to sincerely thank all reviewers for their careful review and constructive suggestions, which helped us greatly improve our paper. Here, we provide a comprehensive overview of the reviewers' feedback and outline our responses accordingly.

---

# Manuscript Strengths

We thank all reviewers for recognizing the following strengths of our manuscript:

* **The CoE method is innovative and quite inspirable:** it provides interesting insights into using LLM hidden states to determine the LLM decision-making correctness and revelations about the LLM thinking system. (Reviewer 2nBf & in4E)

* **The CoE method is highly scalable and efficient to deploy.** (Reviewer in4E)

* **The motivation from intuitive cognitive phenomenon is reasonable**, and the process of making and verifying hypotheses is structured and natural. (Reviewer 2nBf & GkSP)

* **The literature research is comprehensive.** (Reviewer GkSP)

* **The experimental models and datasets are comprehensive.** (Reviewer GkSP)

* **The experimental analysis is detailed and multi-dimensional.** (Reviewer GkSP)


# Concerns and Suggestions

The reviewers also raised some concerns and suggestions, which we have responded to one by one:

* Self-evaluation as the Targeted Topic

  * *Reasonableness of the selected topic (Reviewer GkSP):* We have given clarifications and literature citations and categorized existing methods to argue for the reasonableness of our selected topic.

  * *Open-world generalizability exploration (Reviewer in4E)*: We have added experimental results and demonstrated open-world generalizability.

* More Experiments

  * *Statistic analysis of discrepancies (Reviewer GkSP):* We have given numerical data for the statistical experiment of Figure 2 and pointed out the misunderstanding.

  * *More dataset evaluation (TriviaQA and TruthfulQA) (Reviewer GkSP)*: We have added experimental results.

  * *Robustness analysis on adversarial samples (Reviewer GkSP)*: We have added experimental results.

  * *Model layer analysis (Reviewer in4E)*: We have added experimental results, and pointed out the equivalent implications of Table 1 results for this conclusion.


# Detailed Questions

Regarding some details of our manuscript, some reviewers raised questions and we have explained accordingly:

* *How to obtain the classification threshold? (Reviewer GkSP)*: We have given the calculations as well as further results.

* *Why use the average embedding at layer l? (Reviewer GkSP)*: We have given the explanations with the corresponding literature citation.

* *How to identify the correctness in the evaluation part? (Reviewer in4E)*: We have given examples and pointed out the elaboration of in the corresponding positions of our manuscript (Appendix C.2.3)

# Open Discussion

We also received a discussion of the extended application of our paper from reviewer 2nBf. We have given three possible application scenarios, including the basic data optimization (sampling decoding and preference optimization) as well as the advanced model optimization (improving the training objective) scenarios, and give simple experimental results for the last scenario to validate the future value of our CoE method.

---

**Notably, during the rebuttal phase, we successfully addressed all the reviewers' concerns and did not receive any additional issues.** Finally, We sincerely thank all reviewers and AC for their efforts. We've learned a lot from all the reviews.

Best regards,

Submission255 Authors

---

### Meta-Review · Area_Chair_TXN9 · 2024-12-24

**Metareview:**

## Summary:
The paper proposed the Chain-of-Embedding (CoE), an output-free self-evaluation by large language models (LLMs). The key idea is to analyze LLMs' hidden states during inference across layers, forming a latent "thinking path." CoE investigates the discrepancies in progressive hidden states when LLMs generate correct and incorrect responses to estimate response correctness effectively. This label-free method produces efficient accurate estimation across diverse domains. The efficiency and interpretability of CoE make it suitable for large-scale applications, achieving state-of-the-art results across various datasets and backbone LLMs while offering valuable insights into LLM response correctness.

## Strengths:
1. CoE leverages the internal hidden states' difference between layers for self-evaluation, which provides novel insights into the thinking system of LLMs.
1. It does not require training or ground truths. The computation of the two CoE metrics is scalable for quick feedback in large-scale deployments.
1. CoE offers insights into LLM decision-making by tracing hidden states, providing a transparent and human-like assessment of response correctness.
1. The authors thoroughly verify their proposed metrics across various backbones and datasets, providing detailed analyses in many aspects.

## Weaknesses:
1. Adding discussions about how the discovery can potentially improve the reasoning capability of LLMs can strengthen the paper's contribution and better motivate the study. During the discussion, the authors incorporated the metrics into the training objective and achieved positive results. It would be great to include more complete experiments in the next version.
1. More clarification on the definition of self-evaluation and the motivation of human thinking analogy is needed.
1. Experiments on more diverse tasks, the robustness of the patterns under perturbations, and metrics to determine the threshold were not provided in the original submission. The authors provided comprehensive experiments in the discussion, which well addressed the concerns. These results can make the conclusions more solid and need to be included in the next version.

## Decision:
The authors provided further clarifications and additional experimental results in the rebuttal, as requested by the reviewers. All three reviewers participated in the discussion and confirmed how well their concerns have been addressed by the authors. Two reviewers raised their original ratings and all the reviewers voted for acceptance in the final ratings. The authors did an awesome job in addressing the review comments, especially in providing comprehensive experimental results as solid evidence for their claims. Most reviewers, including the meta-reviewer, agree that the observed CoE patterns and the proposed metrics are interesting and inspirational to the community. To better share this result with the community, the meta-reviewer hereby recommends this paper for acceptance.

**Additional Comments On Reviewer Discussion:**

The authors provided further clarifications and additional experimental results in the rebuttal, as requested by the reviewers. All three reviewers participated in the discussion and confirmed how well their concerns have been addressed by the authors. Two reviewers raised their original ratings and all the reviewers voted for acceptance in the final ratings. The authors did an awesome job in addressing the review comments, especially in providing comprehensive experimental results as solid evidence for their claims.

---

### Decision · Program_Chairs · 2025-01-22

Accept (Poster)